# Complementary CRISPR screen highlights the contrasting role of membrane-bound and soluble ICAM-1 in regulating antigen-specific tumor cell killing by cytotoxic T cells

Ann-Kathrin Herzfeldt[1], Marta Puig Gamez[1], Eva Martin[2], Lukasz Miloslaw Boryn[3], Praveen Baskaran[4], Heinrich J Huber[5], Michael Schuler[2], John E Park[1]*, Lee Kim Swee[1]

[1]Department of Cancer Immunology and Immune Modulation, Boehringer Ingelheim, Biberach an der Riss, Germany; [2]Department of Drug Discovery Sciences, Boehringer Ingelheim, Biberach an der Riss, Germany; [3]Ardigen SA, Kraków, Poland; [4]Department of Global Computational Biology and Digital Sciences, Boehringer Ingelheim, Biberach an der Riss, Germany; [5]Drug Discovery Sciences, Boehringer Ingelheim, Biberach an der Riss, Germany

*For correspondence:
john.park@boehringer-ingelheim.com

**Abstract** Cytotoxic CD8 +T lymphocytes (CTLs) are key players of adaptive anti-tumor immunity based on their ability to specifically recognize and destroy tumor cells. Many cancer immunotherapies rely on unleashing CTL function. However, tumors can evade killing through strategies which are not yet fully elucidated. To provide deeper insight into tumor evasion mechanisms in an antigen-dependent manner, we established a human co-culture system composed of tumor and primary immune cells. Using this system, we systematically investigated intrinsic regulators of tumor resistance by conducting a complementary CRISPR screen approach. By harnessing CRISPR activation (CRISPRa) and CRISPR knockout (KO) technology in parallel, we investigated gene gain-of-function as well as loss-of-function across genes with annotated function in a colon carcinoma cell line. CRISPRa and CRISPR KO screens uncovered 187 and 704 hits, respectively, with 60 gene hits overlapping between both. These data confirmed the role of interferon-γ (IFN-γ), tumor necrosis factor α (TNF-α) and autophagy pathways and uncovered novel genes implicated in tumor resistance to killing. Notably, we discovered that *ILKAP* encoding the integrin-linked kinase-associated serine/threonine phosphatase 2 C, a gene previously unknown to play a role in antigen specific CTL-mediated killing, mediate tumor resistance independently from regulating antigen presentation, IFN-γ or TNF-α responsiveness. Moreover, our work describes the contrasting role of soluble and membrane-bound ICAM-1 in regulating tumor cell killing. The deficiency of membrane-bound ICAM-1 (mICAM-1) or the overexpression of soluble ICAM-1 (sICAM-1) induced resistance to CTL killing, whereas PD-L1 overexpression had no impact. These results highlight the essential role of ICAM-1 at the immunological synapse between tumor and CTL and the antagonist function of sICAM-1.

## Editor's evaluation

This important study uses complementary cutting-edge CRISPR approaches (CRISPR and CRISPRa) to identify novel determinants of cytotoxic CD8 T cell (CTL)-mediated tumor cell killing in vitro. The

Authors use these screens to identify that the integrin-linked kinase ILKAP and the integrin protein ICAM1 both mediate resistance to CTL-mediated killing, leading to a new understanding of how some tumours may evade killing by T cells. The strength of the evidence for these findings is exceptional and backed up by the study of several cancer cell lines as well as human data. This work will be of great interest to tumor immunologists as well as those studying evasion of checkpoint therapy in cancer treatment.

## Introduction

Interactions between tumor cells and the immune system are complex and dynamically regulated. How tumors can acquire resistance to anti-tumor immunity is poorly understood (*Jenkins et al., 2018*; *Schoenfeld and Hellmann, 2020*). A detailed molecular understanding of tumor evasion mechanisms will enable the development of new strategies to exploit the full potential of immunotherapies (*Kalbasi and Ribas, 2020*; *Sambi et al., 2019*; *Sharma et al., 2017*; *Yang, 2015*). Tumor susceptibility to CTL-mediated killing is among others dependent on genetically encoded tumor intrinsic factors (*Kalbasi and Ribas, 2020*; *Sharma et al., 2017*). A series of recent studies have uncovered factors implicated in resistance to CTL-mediated killing through straight forward CRISPR/Cas9 or siRNA-based loss-of-function screens (*Hou et al., 2021*; *Kearney et al., 2018*; *Khandelwal et al., 2015*; *Lawson et al., 2020*; *Manguso et al., 2017*; *Mezzadra et al., 2019*; *Pan et al., 2018*; *Patel et al., 2017*; *Vredevoogd et al., 2021*; *Vredevoogd et al., 2019*; *Young et al., 2020*). Those screens uncovered genes involved in antigen presentation, IFN-γ and TNF-α response pathway as well as autophagy. Tumor cell IFN-γ sensitivity is regulated by the PBAF complex (*Pan et al., 2018*), schlafen 11 (*Mezzadra et al., 2019*) and interaction of the apelin receptor with JAK1 (*Patel et al., 2017*). Maintaining tumor cell fitness after IFN-γ exposure is regulated by the lipid-droplet-related gene (*Fitm2*; *Lawson et al., 2020*). The phosphatase encoded by *Ptpn2* was shown to modulate IFN-γ-mediated effects on antigen presentation and growth (*Manguso et al., 2017*). Despite tumor IFN-γ responsiveness, tumor cell sensitivity to TNF-α influences tumor resistance to CTL attack. Genes such as *Ado* (*Kearney et al., 2018*), *TRAF2* (*Vredevoogd et al., 2019*), *Rb1cc1* (*Young et al., 2020*), *PRMT1* and *RIPK1* (*Hou et al., 2021*) regulate tumor sensitivity to TNF-α. Most of these studies were based on depletion screens which have a lower dynamic range than enrichments screen since genes that confer resistance are depleted. In contrast, in enrichment screens the small number of surviving cells can be enriched by 100-fold or greater reflecting a higher dynamic range of identified gene hits (*Doench, 2018*). One study performed a gain-of-function screen for tumor resistance against T cell cytotoxicity and identified *CD274, MCL1, JUNB*, and *B3GNT2* which enable melanoma cells to evade CTL killing (*Joung et al., 2022*). On the other hand, CRISPR based screens in CD8 +T cells revealed regulators of immune function (*Belk et al., 2022*; *Shifrut et al., 2018*; *Ye et al., 2022*). Using CRISPRa and CRISPR interference (CRISPRi) in parallel enabled functional mapping of gene networks that can modulate cytokine production in primary human T cells (*Schmidt et al., 2022*).

A pan-cancer survey showed that mutations in antigen presentation and interferon signaling pathway were mostly found in melanoma, bladder, gastric and lung cancer (*Budczies et al., 2017*). Although some mechanisms are shared by several cell types, others are cell line specific, likely due to differences in expressed genes and cell biology (*Thelen et al., 2021*). Here, we describe for the first time the combination of a CRISPRa and CRISPR KO screen to study the effect of tumor intrinsic genetic determinants on CTL-mediated killing. Using this approach, we were also able to study regulators that are not expressed endogenously at high levels.

Our CRISPRa and CRISPR KO screens identified 187 and 704 genes implicated in tumor killing respectively, with 60 of them overlapping between both screens. These data confirmed previously identified genes involved in IFN-γ and TNF-α response (e.g. *IFNGR1, JAK2, PTPN2, SOCS1, TNFRSF1A, MAP3K7, CFLAR*), autophagy (e.g. *ATG3, ATG10, ATG12, ATG13*) and others. Our screens uncovered the role of *ILKAP* in protecting tumor cells from antigen specific CTL killing. Moreover, our data show that deletion of mICAM-1 induced stronger resistance compared to PD-L1 overexpression. The overexpression of sICAM-1 induced resistance to killing presumably through inhibition of the interaction between mICAM-1 and LFA-1.

## Results

### In vitro system to investigate genes function in antigen-specific tumor killing

To investigate the effect of intrinsic tumor regulators on antigen-dependent tumor cell killing by CTLs, we established an in vitro tumor cell killing assay (*Figure 1A and B*). To expand CTLs with known antigen specificity, human PBMCs containing CD8 +T cells specific for pp65(495-503) peptide of human cytomegalovirus (CMV) presented in an HLA-A*02:01 restricted manner were stimulated with antigen peptide loaded on MHCI molecules in the presence of IL-2. The stimulation resulted in a 39.4-fold expansion of the antigen-specific CTL population within the PBMCs from 0.64 ± 0.02% to 25.1 ± 2.88% after 8 days (*Figure 1C and D*). CMV-specific CTLs expressed CD25 (19.47 ± 2.85 %), PD-1 (29.49 ± 0.55 %) and LAG-3 (66.69 ± 8.93 %) displaying a more exhausted T cell phenotype after expansion (*Figure 1E*). To assess tumor cell killing, PBMCs containing expanded CTLs were co-cultured with HLA-A*02:01 positive tumor cell lines with different target to effector ratios (T:E). Several tumor cell lines including HCT 116, Panc-1 and UACC-257 were killed by CTLs when loaded with the antigenic peptide (*Figure 1F*). The extent of tumor killing correlated with the ratio of co-cultured PBMCs. *B2M* KO cells were resistant to killing confirming the need of MHCI presentation for specific lysis (*Figure 1G and H*). To activate expression of genes that are not endogenously expressed in cell lines we used the CRISPR dCas9-VPR system. We generated HCT 116 cells which express catalytically deactivated Cas9 (dCas9) fused to the transcriptional activators VP64, p65, and Rta (VPR; *Chavez et al., 2015*) in a stable fashion. To test gene induction, we co-transfected them transiently with crRNAs and trans-activating CRISPR RNA (tracrRNA) to induce the transcription of genes commonly expressed by tumor cells (e.g. *CD274, NT5E*) or genes not expressed by tumor cells such as *CD80*. The expression of *CD274* and *CD80* could be induced and the expression of *NT5E* enhanced (*Figure 1I*). Gene expression reached its maximum after 2 days. After 6 days gene expression levels returned to basal levels. These results show that CRISPR dCas9-VPR system is suitable to induce gene expression of genes that are not endogenously or not naturally (e.g. *CD80*) expressed in this tumor cell line allowing us to survey the function of genes not naturally expressed in our screening cell line.

### Design of a complementary CRISPR activation/KO screen

To identify genes regulating tumor resistance and sensitivity to CTL-mediated killing, we developed a complementary CRISPR screen using CRISPR Cas9 and CRISPR dCas9 methodology (*Figure 2A*). First, *Streptococcus pyogenes* Cas9 and dCas9 single guide RNA (sgRNA) libraries containing 64,556 and 67,833 sgRNAs that target 10,676 and 11,222 genes with annotated function (6 sgRNA per gene) including several non-targeting control sgRNAs were constructed. For the complementary CRISPR screen approach the chemoresistant, microsatellite instability (MSI)-high human colon carcinoma cell line HCT 116 was chosen based on clear correlation between killing and T:E ratio as well as favorable growth properties. Due to higher mutation burden in MSI tumors, it was presumably under high selective pressure in the original patient. Next, tumor cells were engineered by lentiviral transduction to stably express dCas9 and Cas9, respectively. Single cell clones for CRISPRa and CRISPR KO were selected based on their gene editing and activation efficiencies. Cells were then transduced with the respective sgRNA libraries and subjected to geneticin selection for 8 days. Positively selected tumor cells were either left untreated or loaded with CMV antigenic peptide and then exposed to PBMCs containing expanded CTLs at different T:E ratios for 3 days. To achieve moderate killing in CRISPR KO screen a T:E of 2:1 was used, whereas for CRISPRa a T:E of 1:1 was elected to ensure a high selection pressure. The sgRNA library representation in living tumor cells was examined by Next-Generation-Sequencing (NGS). The specificity of sgRNA depletion and enrichment was assessed by comparing different conditions to remove genes controlling cell proliferation and survival (control selection: sgRNA library vs. transduced tumor cells) and to identify genes regulating tumor resistance and sensitivity to antigen-dependent CTL killing (untreated tumor cells with PBMCs vs. antigen loaded tumor cells with PBMCs). To evaluate the efficiency of gene editing or activation in both screens, sgRNA depletion and enrichment in absence of co-culture with PBMC were assessed. As expected, essential genes including genes involved in RNA processing and transport (e.g. *CCA, EEF1A, TGS1*), cell cycle (e.g. *CDK1, SCF, C-MYC, EP300*) and spliceosome (e.g. *PRP2, PRP5, PRP16, PRP22, SNU114, UAP56*) were depleted in the CRIPSR KO screen (*Figure 2B, C, E, F*). Among genes which activation led to decreased fitness we found genes associated with calcium signaling (e.g. *CaV1, CaV2, CaV3, RYR*)

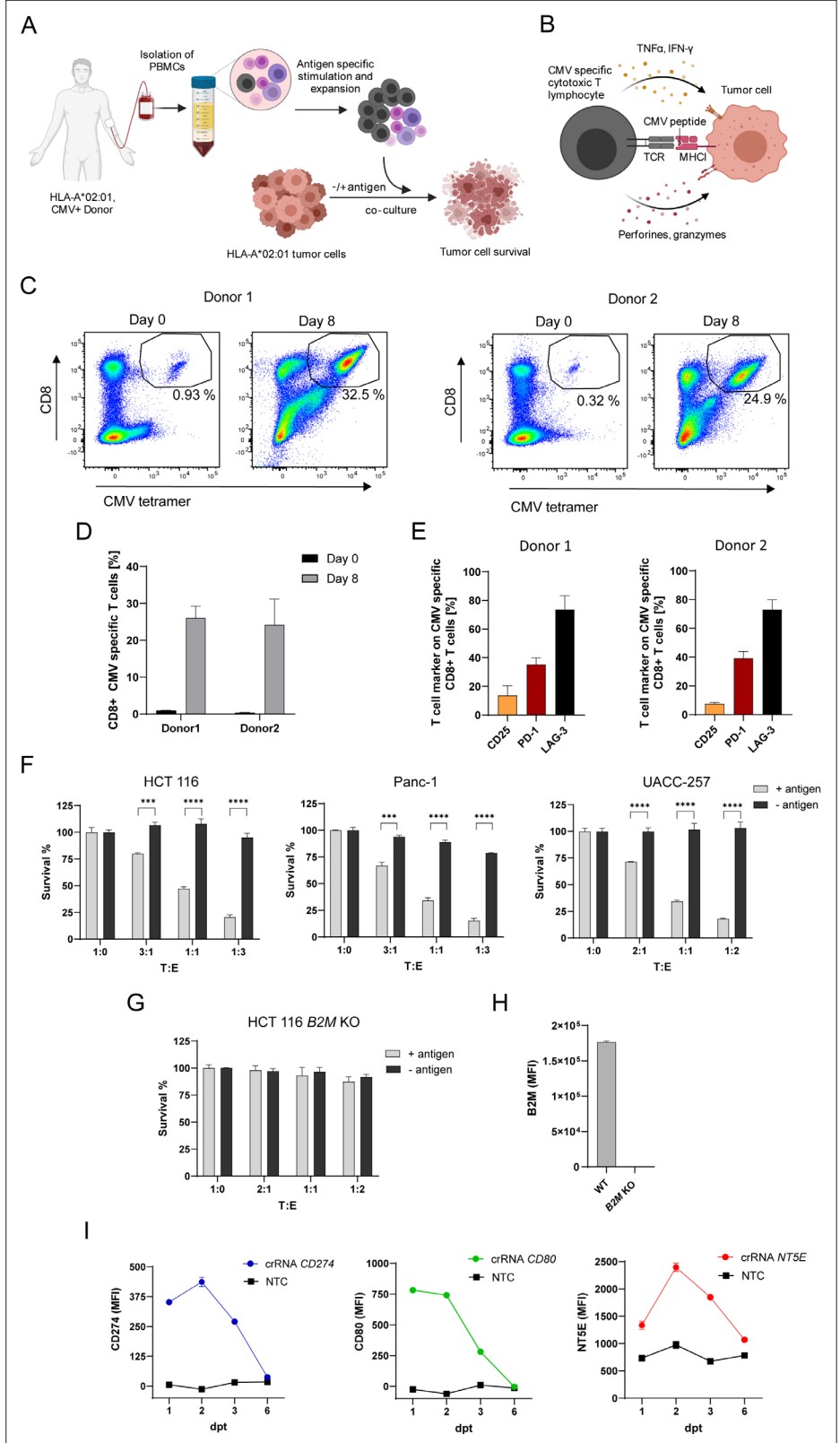

**Figure 1.** In vitro system to investigate genes function in antigen-specific tumor killing. (**A**) Schematic of CMV specific CTL expansion within isolated PBMCs from HLA-A*0201 healthy CMV-seropositive Donors followed by tumor killing assay. Tumor cells either loaded with CMV pp65 antigenic peptide or untreated were co-cultured with PBMCs containing antigen specific CTLs and tumor cell survival was measured using a luminescent cell viability

*Figure 1 continued on next page*

*Figure 1 continued*

assay. (**B**) Schematic of CMV-specific tumor killing by CTLs. CMV-specific CTL recognize CMV antigen presented in an HLA-A*02:01 restricted manner on tumor cells and release cytokines and cytotoxic granules containing perforins and granzymes to specifically kill tumor cells. (**C**) Representative dot plots of CMV pp65495-503 tetramer-positive/CD8 + T cells measured at day 0 and day 8 after stimulation for both Donors used in this study (each n=3). (**D**) Bar graph of acquired frequency of CMV pp65495-503 tetramer-positive/CD8 +T cells (n=3). (**E**) Amount of CD25+, PD-1 +and LAG-3 +CMV specific CD8 + T cells (n=3). (**F**) Cell survival of HCT 116, Panc-1 and UACC-257 after 3 days of co-culturing with different ratios of PBMC containing antigen specific CTLs in antigen presence or absence. Bar graphs show normalized mean ± SD of triplicate representative for three independent experiments. Statistical significance was calculated using two-tailed t tests with adjustments for multiple comparisons (***p<0.001****, p<0.0001). (**G**) Cell survival of HCT 116 *B2M* KO cells assessed with tumor killing assay. Bar graphs show normalized mean ± SD of triplicate representative for two independent experiments. (**H**) Median fluorescence intensity of B2M expression of HCT 116 and B2M KO cells measured with flow cytometry (n=2). (**I**) Mean fluorescence intensities over time of PD-L1, CD80 and NT5E in HCT 116 dCas9 cells after induction of gene expression using CRISPRa compared to non-targeting control (NTC) (n=2).

and ATP-binding cassette transporters (e.g. *ABCA4, ABCB7, ABCB10, ABCC3, ABCC6*) suggesting a disruption of cell homeostasis (*Figure 2B, C, E, F*). The overview of gene coverage per chromosome for both screens confirmed the homogenous distribution of targeted ~10,000 genes throughout the whole genome (*Figure 2D*). Altogether both screens resulted in successful gene disruption or activation throughout the genome regardless of chromosomal location.

## Discovery of genes regulating tumor resistance and sensitivity to CTL killing

To identify tumor intrinsic genetic determinants that modulate resistance and sensitivity to CTL killing, we compared the abundance of sgRNA in tumor cells loaded or not with antigen and co-cultured with PBMCs containing antigen specific CTLs. Tumor cell counts after 3 days co-culture showed that 74% tumor killing was achieved in the CRISPR KO screen and 91% in CRISPRa reflecting moderate and high PBMC selection pressure (*Figure 3A*). With a false discovery rate (FDR) of <5% threshold, our CRISPRa and CRISPR KO screens identified 187 and 704 genes hits respectively with 60 gene hits overlapping between both (*Figure 3B*). The overlap of gene hits found both in CRISPR KO and CRISPRa suggests strong involvement in controlling tumor intrinsic resistance to CTL-mediated killing. Best scoring genes such as *PTPN2, CFLAR, CHD7,* and *ILKAP* induced more sensitivity when depleted and more resistance when activated (*Figure 3C*). On the other hand, *ICAM1* and *JAK2* induced more resistance when depleted and more sensitivity when overexpressed (*Figure 3C*). Additionally, we identified hits specific to CRISPRa screen inducing tumor resistance or sensitivity when overexpressed that were not significantly depleted in CRISPR KO screen, which underlines the importance of examining gene gain-of-function. Analysis of strength and direction of linear relationship of beta score between CRISPR KO and CRISPRa screen gene hits showed a significant negative linear relation in line with the expectation that enriched gene hits in the CRISPRa screen would be depleted in the CRISPR KO screen and vice versa (*Figure 3—figure supplement 1*). Top gene hits identified through both screens involved in for example TNFα signaling were *CFLAR, MAPK1, RIPK1, TNFRSF1A,* and *ICAM1*, highlighting their role in regulating tumor sensitivity to TNF-α-induced cell death. The identification of genes involved in IFN-γ signaling (*PTPN2, SOCS1, STAT1, JAK2*) were consistent with previous findings and validated our complementary CRISPR screen approach (*Lawson et al., 2020*; *Patel et al., 2017*). Furthermore, our data showed additional overlaps with previously performed screens in genes regulating for example autophagy (*PIK3C3, ATG3, ATG10, ATG13*) thus controlling susceptibility to CTL attack (*Lawson et al., 2020*; *Young et al., 2020*). Using gene ontology and pathway analysis, we identified pathways with known function in regulating tumor resistance such as IFN-γ, TNF-α, NF-κβ, autophagy but also novel pathways related to tumor intrinsic immune evasion (*Figure 3D*). In contrast to other studies, enrichment of genes regulating antigen processing and presentation were not found among the top hits in our complementary CRISPR screen presumably due to direct loading of the antigenic peptide on tumor cells. To compare our results to other screens, we examined the intersection between hits from this study and a published tumor resistance core gene set identified through a CRISPR KO screen performed in mouse tumor cells (*Lawson et al., 2020*). Sizeable but incomplete overlap between genes identified through this screen compared to *Lawson et al., 2020*, validate our

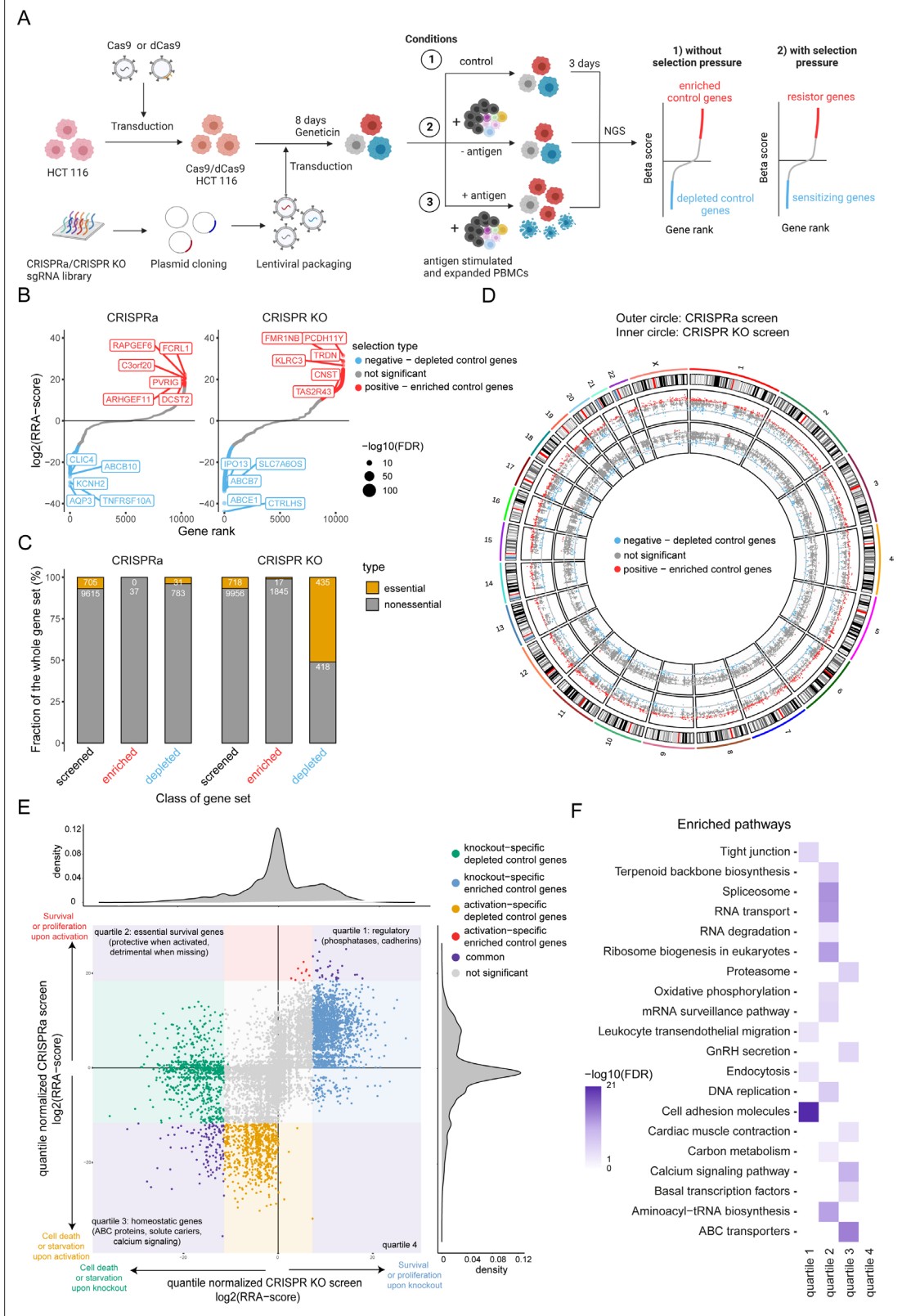

**Figure 2.** Design of a complementary CRISPR activation/CRISPR KO screen. (**A**) Schematic of complementary CRISPR KO/CRISPRa screen setup. HLA-A*0201+ HCT 116 Cas9 or dCas9 colon carcinoma cells were transduced with the respective sgRNA library targeting approx. 10,000 annotated genes. Cells were exposed to PBMCs containing antigen specific CTLs in the presence or absence of CMV antigenic peptide. Control condition was not exposed to PBMCs and antigen. Next-generation sequencing (NGS) was used to determine sgRNA representation of each condition. Each condition

*Figure 2 continued on next page*

*Figure 2 continued*

was performed in triplicate. (**B**) Ranked-ordered, RRA scores (robust ranking aggregation; log2 fold change) for control selection CRISPRa (left) and CRISPR KO (right) screens in absence of PBMCs and antigen. Hits at FDR <2% are highlighted in red (positive selection – enriched control genes) and blue (negative selection – depleted control genes) with the top ten best scoring hits being indicated. (**C**) Enrichment of essential genes (orange; Atlas project - Depmap) as a fraction of gene subset: all screened (black), enriched control genes (red), and depleted control genes (blue) for CRISPRa (left) and CRISPR KO (right) screens. The raw gene counts are indicated in white. (**D**) Overview of gene coverage per chromosome for CRISPR KO (inner circle) and CRISPRa (outer circle); red - enriched control genes, blue - depleted control genes, gray - not significant gene hits. (**E**) Global relation of screened genes between CRISPRa and CRISPR KO assays: purple – common gene hits, red and blue – enriched control gene hits (CRISPRa and CRISPR KO respectively), orange and green - depleted control gene hits (CRISPRa and CRISPR KO respectively). (**F**) Most significant pathways according to KEGG enriched among the significant gene hits of (**E**).

The online version of this article includes the following source data for figure 2:

**Source data 1.** The excel file contains enrichment/depletion scores for each gene, their significance and categorization (control condition: tumor cells only).

approach while demonstrating that it also discovered numerous novel genes (*Figure 3E*). The top 5 ranked genes were indicated in each sector. A key immune evasion mechanism is the loss of TNFα pathway related genes (*Kearney et al., 2018*). TAK1 (*MAP3K7*) is a key regulator of TNFα induced signaling controlling the balance between cell survival and death which was found in our killing screen as well as in other CRIPSR KO screens investigating tumor resistance mechanisms to CTL-mediated killing (*Vredevoogd et al., 2019*; *Young et al., 2020*). Thus, to confirm the role of TNFα signaling in tumor resistance to CTL killing in our model, we assessed tumor cell survival in presence or absence of a TAK1 inhibitor (Takinib). Addition of Takinib significantly enhanced tumor killing in a dose-dependent manner compared to control condition rendering tumor cells more sensitive to TNFα-induced cell death (*Figure 3F*). Taken together, our complementary CRISPR screen identified previously known genes as well as novel gene hits regulating tumor susceptibility to CTL-mediated killing.

## Depletion of *ICAM1* induces tumor resistance to antigen-specific CTL killing

The role of ICAM-1 in the immune response is well documented but its role in regulating anti-tumor response and tumor-CTL interaction remains elusive. Although there are other ICAM family members with overlapping functions and the ability to bind similar ligands (*Binnerts et al., 1994*; *Campanero et al., 1993*; *Casasnovas et al., 1999*), we did not identify other ICAMs in our screen. The most important ICAM-1 ligand for the interaction between CTLs and tumor cells is LFA-1 (*Jenkinson et al., 2005*; *Marlin and Springer, 1987*). LFA-1 is present on the antigen-specific CTLs used in this model (*Figure 4A*). To validate the role of ICAM-1 in controlling tumor cell sensitivity to killing by CTLs, we disrupted *ICAM1* in three tumor cell lines expressing low, medium, and high ICAM-1 (HCT 116, Panc-1 and UACC-257, respectively) using two different sgRNAs. Depletion of ICAM-1 in these cell populations was confirmed by cell surface staining (*Figure 4B*). ICAM-1 deletion led to resistance to CTL killing in all cell lines tested (*Figure 4C*). Resistance could not be attributed to an increase in antigen presentation as HLA-A2 cell surface level was not affected by *ICAM1* depletion (*Figure 4D*). PD-L1 level on the cell surface was increased in UACC-257 cells upon *ICAM1* depletion induced by sgRNA2 (*Figure 5D*). To investigate the role of PD-1-PD-L1 axis in our system, we activated *PDL1* expression by using CRISPRa in tumor cells and measured killing in the presence or absence of Nivolumab (anti-PD-1 antibody) (*Figure 4E, F*). Our results demonstrate that the interaction of PD-1 on antigen-specific CTLs (*Figure 4G*) with PD-L1 had little to no role in the interaction of activated CTLs with tumor cells. PD-1/PD-L1 blockade may rather increase T cell priming and expansion (*Borst et al., 2021*; *Lin et al., 2018*; *Peng et al., 2020*). Altogether these results show that in our system, ICAM-1 plays a crucial role in the productive interaction between tumor and activated CTL and that ICAM-1 depletion has more effect than PD-1 overexpression in inducing killing resistance.

## ICAM-1 isoforms differently regulate antigen specific tumor cell killing by CTLs

Multiple isoforms of ICAM-1 exist including secreted variants (*Ramos et al., 2014*; *Seth et al., 1991*; *Wakatsuki et al., 1995*). Secreted ICAM-1 may in fact function as LFA-1 antagonist (*Meyer et al., 1995*) altogether mimicking ICAM-1 deficiency by disrupting mICAM-1/LFA1 interaction. Shedding of

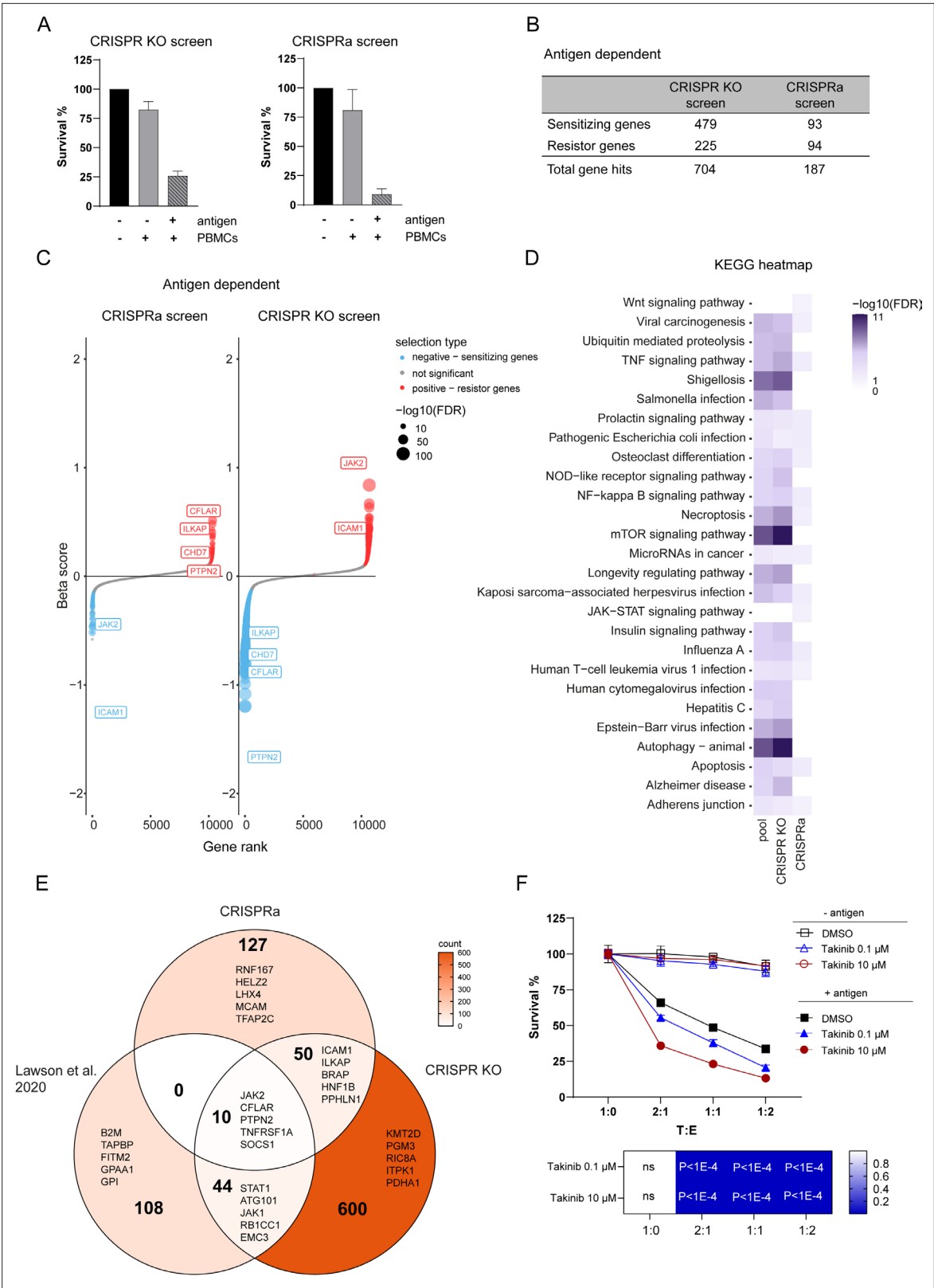

**Figure 3.** Discovery of genes regulating tumor resistance and sensitivity to CTL killing. (**A**) Cell survival after co-culturing with PBMCs containing antigen specific CTLs for 3 days normalized to tumor cells not exposed to PBMCs and antigen for CRISPR KO (left) and CRISPRa (right) screen. (**B**) Table displaying the numbers of gene hits specific for antigen-dependent setup identified by CRISPR KO and CRISPRa screen. (**C**) Ranked-ordered, beta-scores for antigen-dependent screen setup (CRISPRa – left; CRISPR KO – right). The top best scoring overlapping gene hits between CRISPR KO

*Figure 3 continued*

and CRISPRa screen are indicated. Hits at FDR <5% are highlighted in red (positive selection - resistor genes) and blue (negative selection - sensitizing genes). (**D**) KEGG pathway enrichments for top 15 best scoring pathways in CRISPR KO, CRISPRa or pooled screen hits represented as heatmap: white – not statically significant (FDR corrected hypergeometric overrepresentation test). (**E**) Venn diagram displaying intersection of CRISPRa screen gene hits, CRISPR KO screen gene hits and previously published tumor resistance core gene data set of *Lawson et al., 2020*. Top 5 ranked genes were indicated in each sector. (**F**) Tumor killing assay in the presence of different concentrations of TAK1 inhibitor (Takinib) as indicated or DMSO control and cell survival was measured after 3 days (top). Bar graphs show normalized mean ± SD in triplicate representative for two independent experiments. Two-way ANOVA corrected for multiple comparison according to Dunnett was used to determine statistical significance (bottom) (ns: not significant).

The online version of this article includes the following source data and figure supplement(s) for figure 3:

**Source data 1.** Complete list of screened genes containing beta-scores, FDR and specificity.

**Figure supplement 1.** Correlation between CRISPR KO and CRISPRa screen gene hits within certain pathways.

ICAM-1 from the cell surface is mediated by several proteases including MMP-9 (*Fiore et al., 2002*). The substitution of proline in position 404 with glutamic acid (P404E) inhibited shedding of ICAM-1 in the presence of MMP-9 without affecting mICAM-1 levels (*Fiore et al., 2002*). ICAM-1 cleavage process is regulated by multiple kinases acting through specific tyrosine residues Y474 and Y485 within the cytoplasmic region of ICAM-1 (*Tsakadze et al., 2004*). In order to investigate the role of ICAM-1 isoforms and such mutants important for ICAM-1 shedding, we transfected *ICAM1* KO or WT cells with plasmids encoding for ICAM-1 variants (*Figure 5A*) and investigated tumor cell killing by CTLs. To monitor transfection efficacy and kinetics of tumor killing, all plasmids contained enhanced GFP (eGFP) (*Figure 5B, C*). The fraction of eGFP + cells after transfection was similar between all ICAM-1 variants reflecting equal transfection efficiency (*Figure 5C*). Detection of ICAM-1 variants via cell surface staining against N-terminal DYKDDDDK Tag (flag-tag) showed differential levels of ICAM-1 in the plasma membrane upon transfection (*Figure 5D*). Flag-tag levels of mutated *ICAM1* (P404E), ICAM-1 lacking cytoplasmic tail (*ICAM1-ΔC*) and GPI-anchored ICAM-1 (*ICAM1-ΔTM-ΔC-GPI*) were comparable to full length *ICAM1*. Expression levels of mutant *ICAM1* Y474A+Y485 A were lower compared to other ICAM-1 variants as measured by Flag-tag. Mutant versions of ICAM-1, Y474A+Y485 A and P404E, were previously described to inhibit proteolytic cleavage and subsequent shedding of ICAM-1 in other cell types (*Fiore et al., 2002*; *Tsakadze et al., 2004*). In our model, neither mICAM-1 levels (*Figure 5D*) nor secreted amounts of sICAM-1 (*Figure 5E*) were altered after transfection compared to full length ICAM-1. These results indicate that these mutations are not relevant for ICAM-1 cleavage under these conditions in HCT 116 cells.

Transfection of *sICAM1* in *ICAM1* KO cells resulted in no detectable flag-tag expression on the cell surface, but enhanced sICAM-1 levels in the supernatant 5.21±0.42 fold (*Figure 5E*). Additionally, reintroduction of full length *ICAM1* in *ICAM1* KO resulted in 2.21±0.11 fold higher sICAM-1 levels. Inversely, *ICAM1* KO cells secrete 4-fold less compared to WT cells (*Figure 5E*). Levels of sICAM-1 in the supernatants of WT cells transfected with *sICAM1* were 2.39±0.19 fold higher than in control WT cells (*Figure 5E*).

Finally, we co-cultured tumor cells transfected with ICAM-1 variants with PBMCs containing expanded antigen specific CTLs and monitored tumor cell killing over time. The expression of full-length *ICAM1* rescued antigen-specific tumor cell killing by CTLs in *ICAM1* KO cells confirming the important role of ICAM-1 in controlling CTL-mediated killing (*Figure 6A*). We also tested two computationally mapped potential isoforms of ICAM-1 (source UniProt) which proved neither detectable on the cell surface nor in the supernatant and therefore, as expected, had no effect on tumor killing (data not shown). The mutant *ICAM1* P404E rescued tumor killing by CTLs to similar extent as full length *ICAM1,* whereas no rescue could be detected upon transfection with *ICAM1* Y474A+Y485 A (*Figure 6A*). These data emphasize the importance of the ratio of mICAM-1 and sICAM-1 for the productive interaction between tumor cells and CTLs. No significant change in killing could be detected upon expression of *sICAM1* in *ICAM1* KO cells. Overexpression of sICAM-1 in WT cells had no impact on mICAM-1 expression (*Figure 6—figure supplement 1*) but protected cells from CTL killing, possibly due to interference of sICAM-1 with mICAM-1/LFA-1 interaction (*Figure 6B*). To investigate the cell extrinsic effect of sICAM-1 on killing, tumor killing was investigated in the presence of conditioned media containing external sICAM-1 and untreated tumor cells. Addition of medium harvested from cells overexpressing sICAM-1 (*Figure 6—figure supplement 2A, B*) or medium with recombinant sICAM-1 (*Figure 6—figure supplement 3A, B*) had no effect on tumor killing. These

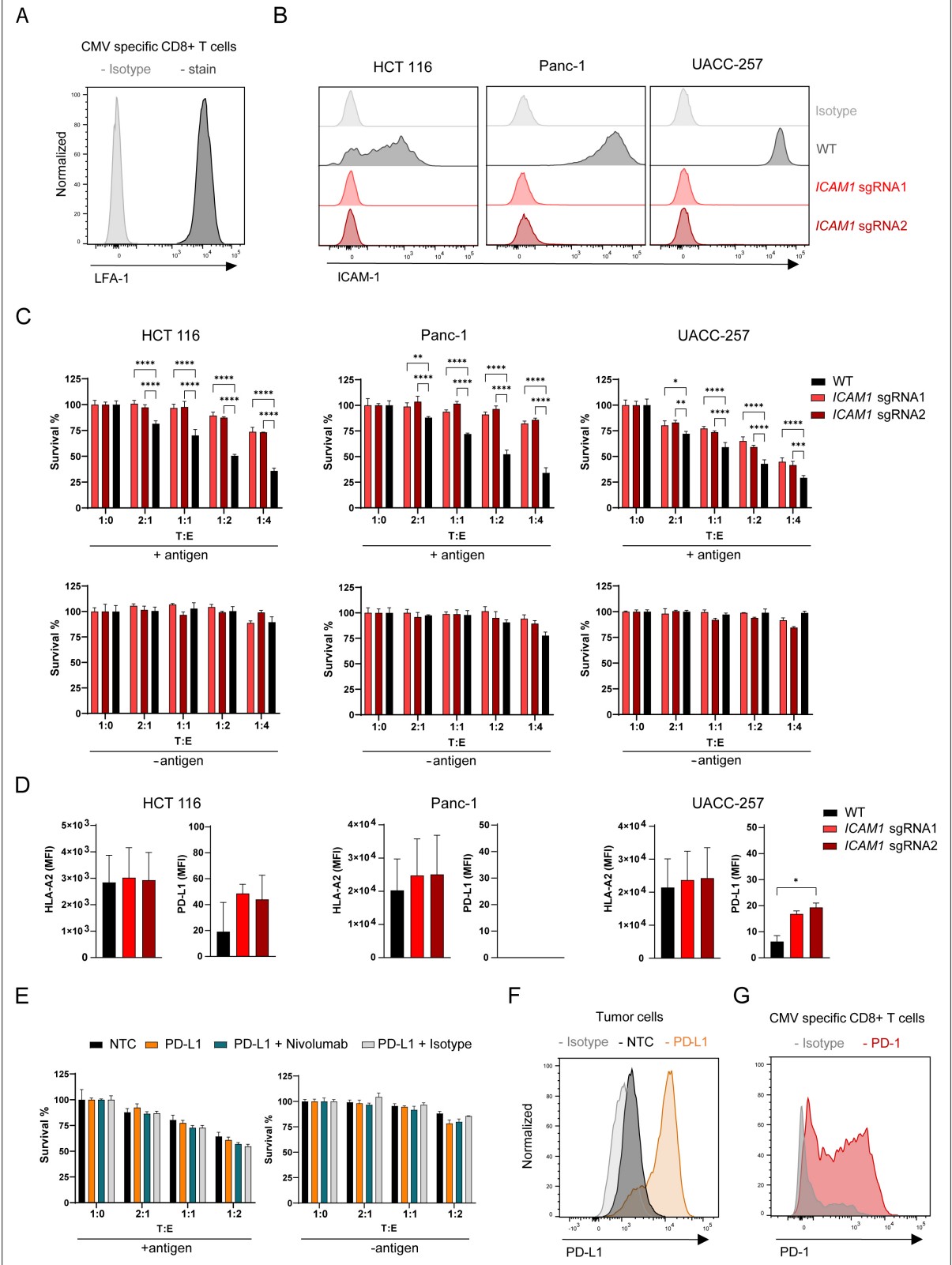

**Figure 4.** Depletion of *ICAM1* induces tumor resistance to antigen-specific CTL killing. (**A**) LFA-1 cell surface expression of CMV-specific CD8 + T cells measured by flow cytometry displayed as histogram (n=2). (**B**) Histograms showing ICAM-1 levels of HCT 116, Panc-1 and UACC-257 cell lines and respective KO pools after fluorescence activated cell sorting (n=3). (**C**) Cell survival of antigen loaded and untreated HCT 116, Panc-1, UACC-257 cells and *ICAM1* KO pools using CRISPR KO and 2 sgRNAs cells against CTL killing after 3 days of co-culturing with different ratios of PBMCs containing

*Figure 4 continued on next page*

Figure 4 continued

antigen specific CTLs. Bar graphs show normalized mean ± SD of triplicate representative for two (Panc-1, UACC-257) or three (HCT-116) independent experiments. Two-way ANOVA corrected for multiple comparison according to Dunnett was used to determine statistical significance (*p<0.05, **p<0.01, ***p<0.001, ****p<0.0001). (D) Mean fluorescence intensities of HLA-A2 and PD-L-1 on the cell surface of HCT 116 WT or HCT 116 *ICAM1* KO cells. Bar graphs show mean ± s.e.m (n=2). Unpaired two-tailed *t* test was used to determine statistical significance (*p<0.05). (E) Cell survival of untreated or antigen loaded HCT 116 and HCT 116 PD-L1 cells in the presence of Nivolumab or isotype with different ratios of PBMCs containing antigen-specific CTLs. Bar graphs show normalized mean ± s.e.m. in triplicate representative for two independent experiments. (F) Representative histogram of CRISPRa-induced *PDL1* expression in HCT 116 cells (n=2). NTC = non-targeting control. (G) Representative histogram of PD-1 expression of stimulated CMV-specific CTLs (n=3).

results demonstrate that cells must produce sICAM-1 to be protected from CTL killing. Truncation of cytoplasmic tail of ICAM-1 (*ICAM1-ΔC*) did not alter rescue of tumor cell killing compared to full-length *ICAM1* (*Figure 6C*). However, *ICAM1-ΔTM-ΔC-GPI* was not as efficient as full-length *ICAM1* in rescuing tumor killing (*Figure 6C*).

In summary, mICAM-1 and sICAM-1 play opposite roles in the interaction of CTL with tumor cells. mICAM-1 promotes tumor killing by CTL whereas sICAM-1 increases resistance.

## Expression of *ICAM1* and ICAM-1 cleavage related metalloproteases is upregulated in human cancers and associated with poor clinical outcome

ICAM-1 is constitutively expressed and up-regulated by inflammatory activation such as stimulation by TNF-α or IFN-γ (*Becker et al., 1991*; *Figenschau et al., 2018*; *Ramos et al., 2014*). To test induction of mICAM-1 expression and sICAM-1 release, we stimulated various tumor cell lines with TNF-α, IFN-γ or the combination of both. Both TNF-α and IFN-γ enhanced mICAM-1 expression and induced release of sICAM-1 in all cell lines tested suggesting this mechanism in generalizable across different cancer types (*Figure 7A and B*). The release of sICAM-1 induced by the combination of both was higher than that induced by the individual cytokines (*Figure 7B*). The soluble form of ICAM-1 is generated by alternative splicing of *ICAM1* or proteolytic cleavage of mICAM-1 through human neutrophil elastase, cathepsin G, MMP-9, ADAM10 and ADAM17 (*Fiore et al., 2002*; *Morsing et al., 2021*; *Robledo et al., 2003*; *Tsakadze et al., 2006*; *Wakatsuki et al., 1995*). To evaluate the expression of *ICAM1* and ICAM-1 cleavage related proteases, we analyzed gene expression of 22 human cancers obtained from The Cancer Genome Atlas (TCGA) and Genotype-Tissue Expression Portal (GTEx). All normal healthy tissue types analyzed expressed *ICAM1* at varying basal levels (*Figure 7C*). In 12 human cancers it was significantly upregulated compared to normal tissue. Moreover, *MMP9* expression was elevated in all tumor types compared to normal (*Figure 7D*). In some tumor types expression of *ADAM10* and *ADAM17* was increased compared to normal tissue. Expression of *ELANE* and *CTSG* was lower compared to normal tissue. Next, we sought to evaluate whether the expression of *ICAM1* and ICAM-1 cleavage related proteases is associated with clinical outcome. In this analysis, patients were categorized into 'high' and 'low' groups according to the highest and the lowest quartiles of each individual gene expression. We found high expression of *ICAM1* and high expression of *MMP9* was related to shorter survival in glioblastoma multiforme patients (*Figure 7E*). Moreover, high expression of *ICAM1* and high expression of *ADAM10* or *ADAM17* was associated with poor clinical outcome in pancreatic adenocarcinoma patients (*Figure 7E*). The expression of the combination of ICAM1 and protease have a worse impact on survival than each gene alone (*Figure 7—figure supplement 1*). Collectively, expression of *ICAM1* and ICAM-1 cleavage related metalloproteinases is elevated in various human cancers. Moreover, high co-expression of *ICAM1* and *MMP9, ADAM10,* or *ADAM17* is associated with poor clinical outcome. Altogether, our data suggest that CTL-mediated tumor cell killing is modulated by mICAM-1 level and release of sICAM-1 (*Figure 7F*). While ICAM-1 contributes to the formation of a productive immunological synapse leading to tumor killing, its absence or release of sICAM-1 inhibits tumor cell killing.

## Depletion of *ILKAP* promotes antigen-specific CTL-mediated tumor cell killing

ILKAP is a protein serine/threonine phosphatase of the PP2C family linked to cancer through phosphorylation of integrin-linked kinase (ILK) thereby modulating downstream integrin signaling.

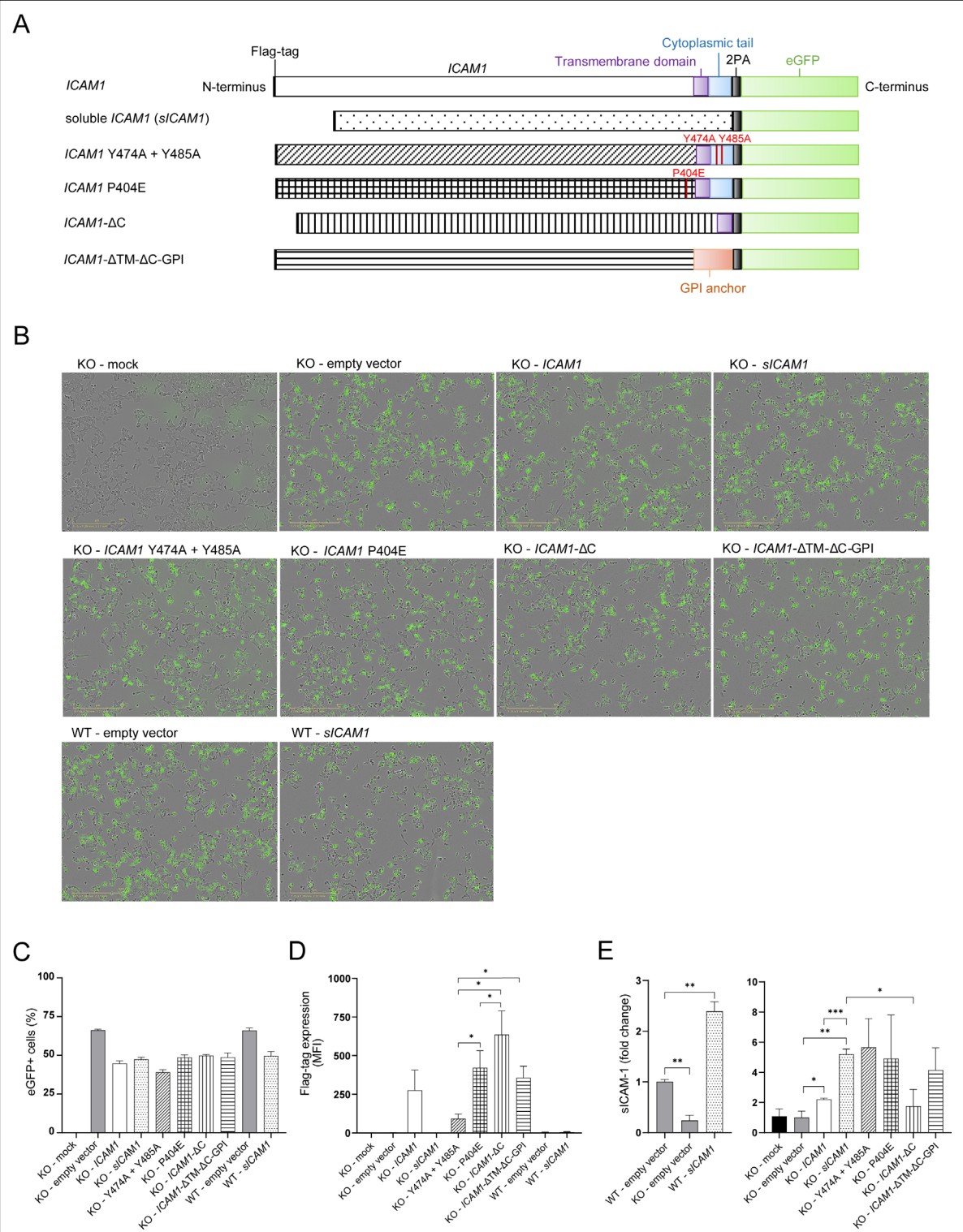

**Figure 5.** Design and expression of different ICAM-1 isoform eGFP-plasmids. (**A**) Design of different ICAM1 isoforms carrying eGFP-plasmids. (**B**) Representative pictures of HCT 116 or *ICAM1* KO cells transfected with ICAM-1-eGFP-plasmids. Pictures were obtained 20 hr after transfection with a 10 x objective using phase contrast channel as well as the green fluorescent channel (n=3). Scale bars, 400 μm. (**C**) eGFP + cells one day post transfection (dpt) measured by flow cytometry. Bar graphs show mean frequency ± s.e.m. (n=3). (**D**) Flag-tag level on the cell surface after 1 day of transfected cells measured by flow cytometry. Bar graphs show mean fluorescent intensity ± s.e.m. (n=3). (**E**) Fold change of sICAM-1 in the supernatant of transfected cells compared to WT (left) or KO (right) measured by IQELISA. Bar graphs show mean ± s.e.m. (n=3.). Two-tailed *t* tests with adjustments for multiple comparisons were performed (*p<0.05, **p<0.01, ***p<0.001).

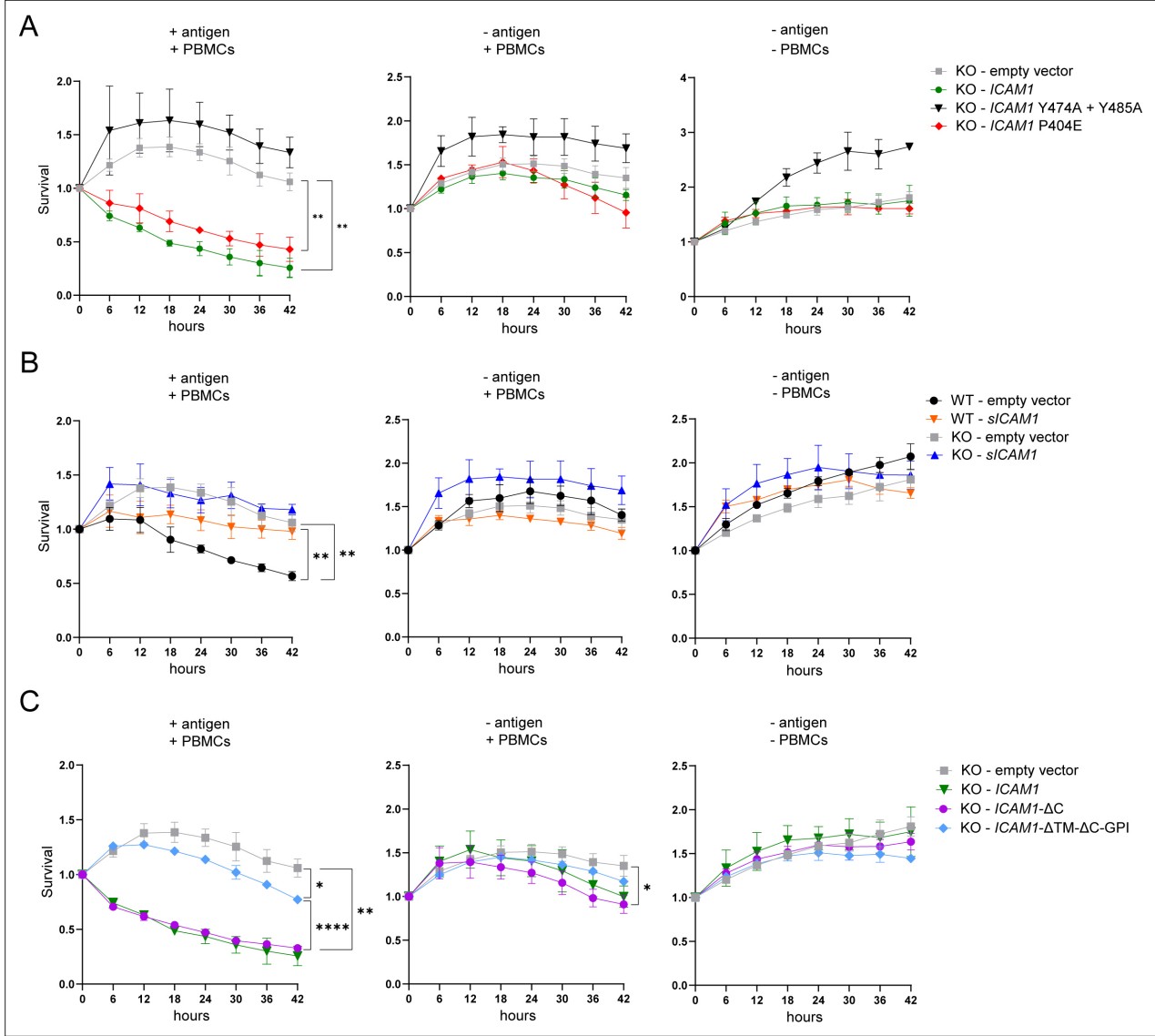

**Figure 6.** ICAM-1 isoforms differently regulate antigen-specific tumor cell killing by CTLs. (**A**) Real time kinetic of tumor cell killing by PBMCs with T:E ratio of 1:4. HCT 116 *ICAM1* KO cells were transfected with empty vector (gray), *ICAM1* (green), *ICAM1* Y474A+Y485 A (black) or *ICAM1* P404E (red). (**B**) Real-time kinetic of tumor cell killing by PBMCs with T:E ratio of 1:4. WT or HCT 116 *ICAM1* KO cells were transfected with empty vector (WT – black; KO – gray) or *sICAM1* (WT – orange; KO – blue). (**C**) Real-time kinetic of tumor cell killing by PBMCs with T:E ratio of 1:4. HCT 116 *ICAM1* KO cells were transfected with empty vector (gray), *ICAM1- ΔC* (purple), *ICAM1-ΔTM-ΔC-GPI* (light blue). Cell survival was determined counting green objects every 6 hours by using the IncuCyte system and normalized to timepoint zero. Conditions were performed in triplicate and four pictures of each triplicate were used for analysis (in total 12). Line graphs show mean ± SD for each timepoint representative for at least two independent experiments. Two-way ANOVA with Geisser-Greenhouse correction was used to determine statistical significance of each timepoint. Depicted stars represent statistical significance for t=42 hr (*$p < 0.05$, **$p < 0.01$, ***$p < 0.001$, **** $p < 0.0001$).

The online version of this article includes the following source data and figure supplement(s) for figure 6:

**Source data 1.** Sequences of ICAM-1 isoform eGFP-plasmids.

**Figure supplement 1.** mICAM-1 levels measured by flow cytometry of HCT 116 cells transfected with full length ICAM-1, sICAM-1 or empty vector as control.

**Figure supplement 2.** Media enriched with sICAM1 from stimulated Panc-1 cells does not protect HCT116 cells from CTL killing.

**Figure supplement 3.** Addition of recombinant sICAM1 in media does not protect HCT 116 cells from CTL killing.

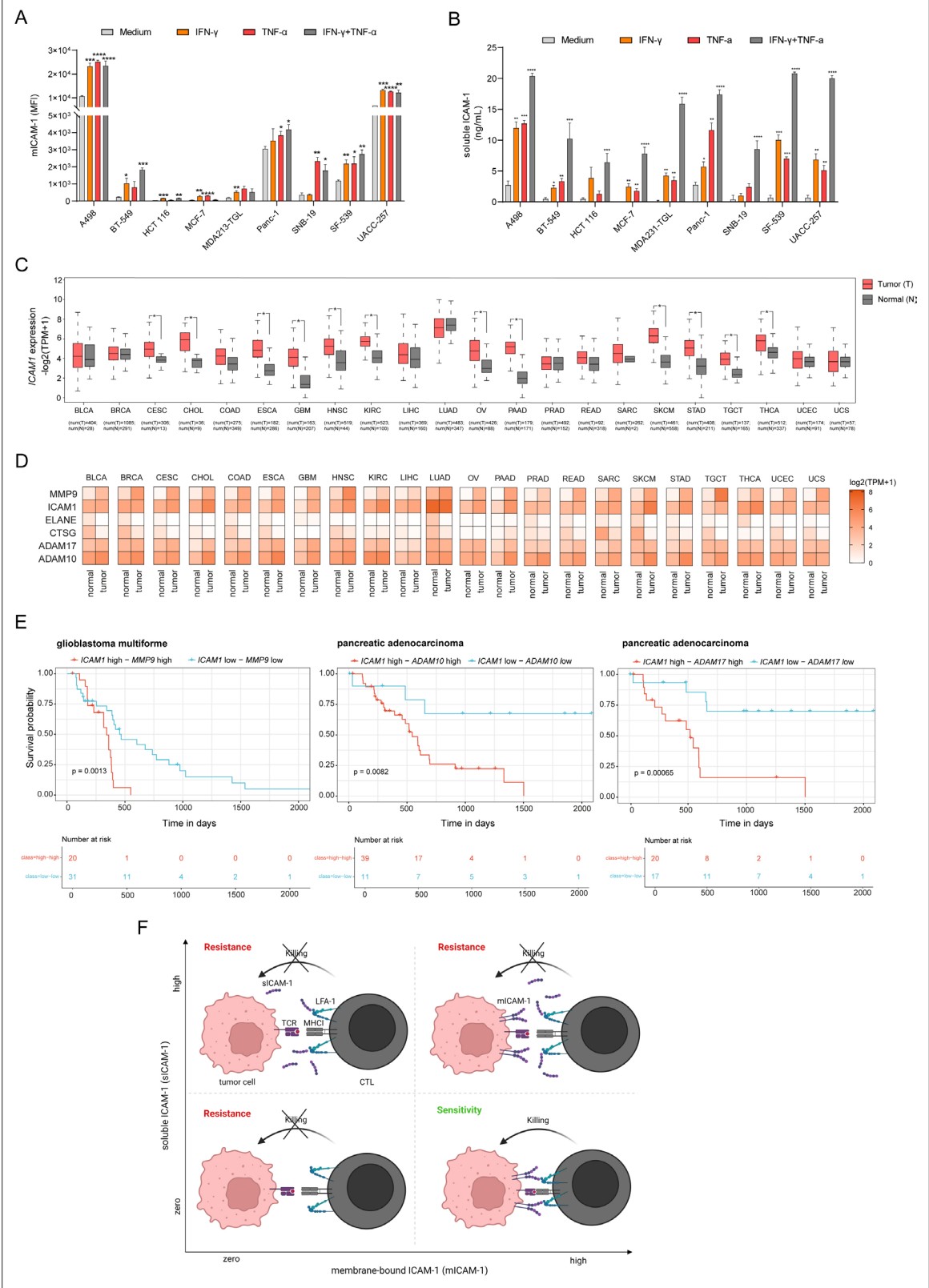

**Figure 7.** Expression of *ICAM1* and ICAM-1 cleavage related metalloproteases is upregulated in human cancers and associated with poor clinical outcome. (**A**) Membrane-bound ICAM-1 (mICAM-1) on the cell surface and (**B**) soluble ICAM-1 in the supernatant of untreated or stimulated cells with 100 ng/mL IFN-γ, 20 ng/mL TNF-α or both. Bar graphs show normalized mean ± SD of triplicate for each condition. Two-tailed *t* tests with adjustments for multiple comparisons were performed (*p<0.05, **p<0.01, ***p<0.001, **** p<0.0001). (**C**) *ICAM1* expression in normal (N) or tumor tissue (T) of 22

*Figure 7 continued on next page*

*Figure 7 continued*

different human cancers. Number of samples used for analysis as indicated. (**D**) Heatmaps showing expression of *ICAM1* and ICAM-1 cleavage related proteases *MMP9, ELANE, CTSG, ADAM10,* and *ADAM17* in normal or tumor tissue of 22 different cancer types. Expression data were obtained using GEPIA. (**E**) Kaplan–Meier survival plots of patient overall survival with the expression of *ICAM1* and *MMP9* (left), *ICAM1* and *ADAM10* (middle), *ICAM1* and *ADAM17* (right). Patients were categorized into 'high' and 'low' groups according to the highest and the lowest quartiles of each individual gene expression. Data were obtained from TCGA and GTEx. (**E**) Schematic describing the effect on tumor killing by mICAM-1 and sICAM-1. More details see text.

The online version of this article includes the following figure supplement(s) for figure 7:

**Figure supplement 1.** Kaplan–Meier survival plots of patient overall survival with the expression of each gene alone.

However, its role in antigen recognition and antigen specific killing has not been characterized. To validate the role of ILKAP in antigen-dependent tumor killing by CTLs, we disrupted gene expression with multiple sgRNAs in HCT 116 and Panc-1 cell lines. The depletion of *ILKAP* induced increased tumor sensitivity to antigen-specific CTL killing in both cell lines which correlated with remaining expression (*Figure 8A–C*). The effect of *ILKAP* depletion and basal expression in Panc-1 cells on CTL-mediated tumor killing was more moderate compared to HCT 116 cells. To investigate if ILKAP induces tumor resistance to CTL killing through a mechanism dependent on regulating IFN-γ or TNFα sensitivity, we stimulated *ILKAP* KO HCT 116 clone with IFN-γ or TNFα. No significant difference in cell death between *ILKAP* KO and control cells upon IFN-γ or TNFα stimulation could be detected (*Figure 8D*). Next, to explore if ILKAP regulates antigen presentation, cell adhesion or PD-L1 expression, we measured cell surface levels of HLA-A2, ICAM-1, and PD-L1. Upregulation of HLA-A2, ICAM-1 and PD-L1 was similar between *ILKAP* KO and control cells upon INF-γ or TNF-α stimulation (*Figure 8E*). Interestingly, *ILKAP* KO cells showed an enhanced basal level of ICAM-1 compared to control cells, whereas PD-L1 and HLA-A2 levels were similar (*Figure 8F*). To further clarify the connection between ILKAP and ICAM-1, ILKAP and a catalytic inactive mutant of ILKAP (H154D) unable to inhibit ILK activity (*Leung-Hagesteijn et al., 2001*) were overexpressed in tumor cells and mICAM-1 and sICAM-1 levels were measured. Overexpression of both variants was confirmed by western blot (*Figure 8G*). We observed a non-significant decrease of mICAM-1 and sICAM-1 following transient transfection of ILKAP but not catalytic dead ILKAP (H154D) (*Figure 8H and I*).

Taken together, these results show that *ILKAP* deletion enhances tumor killing independently of increasing sensitivity to IFNγ or TNF. *ILKAP* KO led to concomitant increase in mICAM-1 but *ILKAP* overexpression had minor impact on surface ICAM-1 expression. The characterization of the potential regulation of ICAM-1 by ILKAP would need further investigation.

## Discussion

We developed a complementary CRISPR screen to identify tumor intrinsic genetic determinants that control tumor susceptibility to CTL-mediated killing. In contrast to previous studies, we combined a CRISPRa screen with a CRISPR KO screen to study upregulation of genes that are not expressed endogenously at high levels. In line with previously published CRISPR KO screens in mouse and human tumor cells, we identified genes involved in autophagy, IFN-γ and TNF-α signaling pathway (*Kearney et al., 2018*; *Lawson et al., 2020*; *Patel et al., 2017*; *Vredevoogd et al., 2019*). Due to the external loading of tumor cells with the respective antigenic peptide, tumor intrinsic antigen processing and presentation pathways are not detectable allowing us to map gene hits besides this tumor evasion strategy.

Our approach uncovered *ILKAP* as novel regulator of tumor sensitivity to CTL killing. ILKAP was first identified in a yeast two-hybrid screen associated with Integrin-linked kinase 1 (ILK1) and shown to negatively regulate ILK1 activity thereby targeting ILK1 signaling components of Wnt pathway (*Leung-Hagesteijn et al., 2001*). In the context of cancer, ILKAP was described to regulate the susceptibility of ovarian tumor cells to cisplatin, a platinum-based anti-cancer drug (*Lorenzato et al., 2016*), but never associated with antigen-specific tumor killing by CTLs. Our screens showed that depletion of ILKAP leads to more tumor killing and activation of *ILKAP* expression to more resistance to CTL killing. Upon *ILKAP* KO, we found elevated basal ICAM-1 cell surface levels. It was previously shown that ILK regulates ICAM-1 expression via NF-κB signaling (*Lee et al., 2006*). Since we observed a trend towards decreased levels of ICAM-1 upon overexpression of ILKAP but not with the catalytic

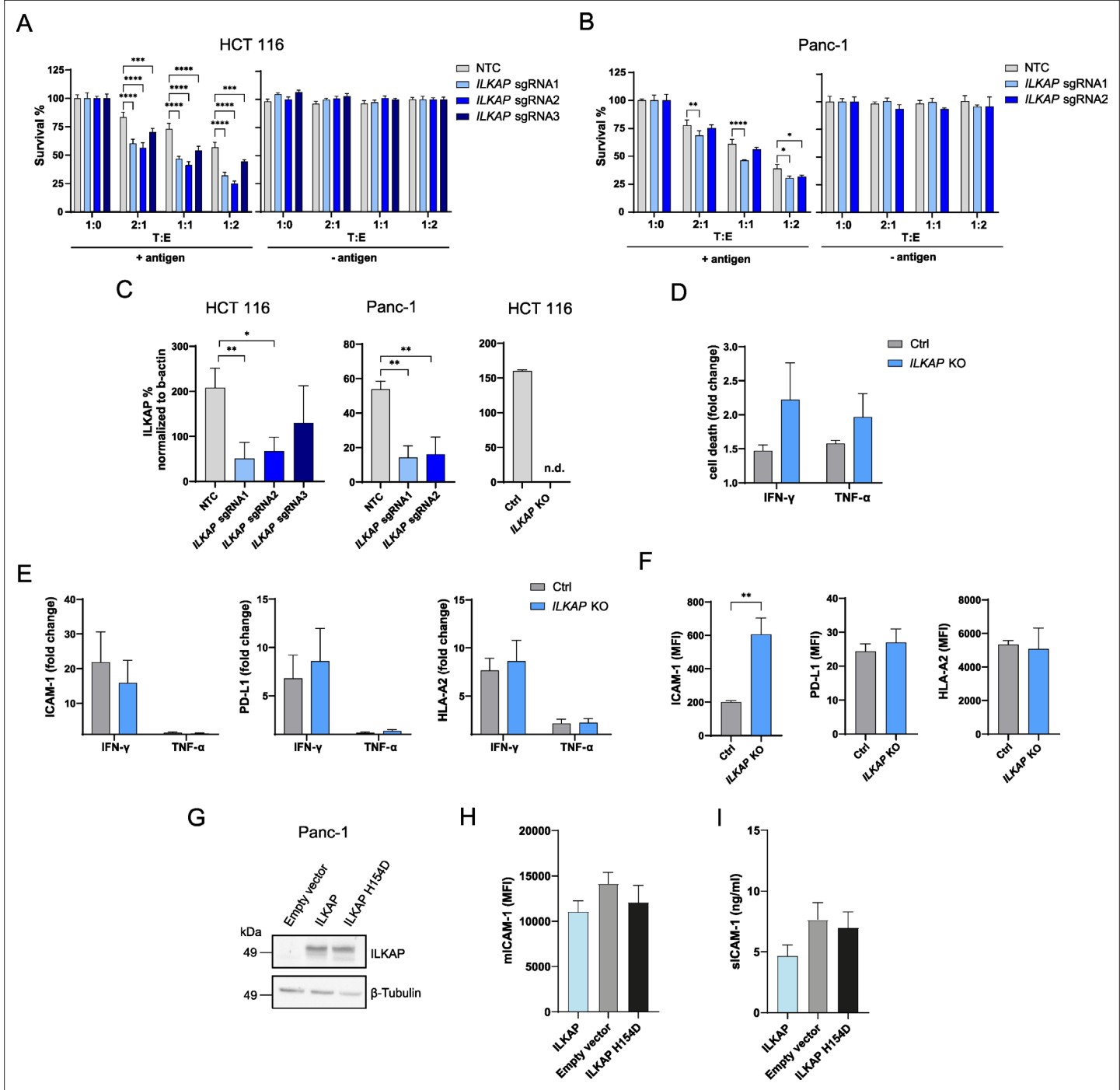

**Figure 8.** Depletion of *ILKAP* promotes antigen-specific CTL-mediated tumor cell killing. (**A**) Cell survival of antigen loaded and untreated HCT 116 WT or *ILKAP* KO cells using 3 sgRNAs after 3 days of co-culturing with different ratios of PBMCs containing antigen specific CTLs. Bar graphs show normalized mean ± SD of triplicate representative of three independent experiments. (**B**) Cell survival of antigen loaded and untreated Panc-1 WT or *ILKAP* KO cells using 2 sgRNAs after 3 days of co-culturing with different ratios PBMCs containing antigen-specific CTLs. Bar graphs show normalized mean ± SD of triplicate representative of three independent experiments (**C**) ILKAP protein levels normalized to β-actin determined by Simple Western system. Bar graphs show normalized mean ± SD (n=3). (n.d. – not detectable). (**D**) Cell death of HCT 116 WT or *ILKAP* KO cells untreated or treated with 100 ng/mL IFN-γ or 40 ng/mL TNF-α determined with live/dead staining (FVS780) using flow cytometry. Bar graphs show mean ± s.e.m (n=3). (**E**) Fold change of HLA-A2, ICAM-1 and PD-L1 cell surface levels after treatment with 100 ng/mL IFN-γ or 40 ng/mL TNF-α of WT or HCT 116 *ILKAP* KO cells. Bar graphs show mean ± s.e.m (n=3). (**F**) Mean fluorescence intensities (MFI) of HLA-A2, ICAM-1 and PD-L1 of HCT 116 WT or HCT 116 *ILKAP* KO cells. Bar graphs show mean ± s.e.m (n=3). (**G**) ILKAP protein levels of Panc-1 cells transiently transfected with ILKAP or ILKAP H154D assessed by western blot. (**H**) Level of mICAM-1 measured with flow cytometry and (**I**) secreted sICAM-1 levels determined by ELISA of transiently transfected Panc-1

*Figure 8 continued on next page*

*Figure 8 continued*

cells. For (**A**) and (**B**), two-way ANOVA corrected for multiple comparison according to Dunnett was used to determine statistical significance (*p<0.05, **p<0.01, ***p<0.001****, p<0.0001). Two-tailed *t* tests with adjustments for multiple comparisons were performed (**D and E**). For (**C and F**) unpaired two-tailed *t* test was used to determine statistical significance (*p<0.05, **p<0.01).

inactive mutant ILKAP (H154), we suggest that ILKAP regulates ICAM-1 basal levels through modulating ILK signaling. The minor impact of overexpression compared to deletion could be explained by already high level of ILKAP phosphatase activity in untransfected cells that may not be dramatically increased upon overexpression. Stimulation of ILKAP KO cells with IFN-γ and TNFα revealed that ILKAP-mediated tumor protection against CTL killing is independent from controlling INF-γ or TNFα sensitivity, changing PD-L1 levels and regulating antigen presentation. Further studies are needed to investigate how ILKAP controls tumor killing by CTLs.

Furthermore, our complementary CRISPR screen showed that activation of *ICAM1* expression enhanced tumor killing by CTLs and depletion attenuated CTL killing. ICAM-1 plays several roles in the immune system including cellular adhesion, inflammation, wound healing, T cell activation and leukocyte recruitment (*Bui et al., 2020*). Importantly, surface ICAM-1 binds to LFA-1 on T cells and contribute to the formation of an immunological synapse between target cells and CTL during killing (*Anikeeva et al., 2005*; *Franciszkiewicz et al., 2013*) as well as antigen presenting cell and T cell during priming (*Hartman et al., 2009*; *Scholer et al., 2008*). Interestingly, the absence of ICAM-1 on tumor cells had a stronger negative impact of tumor killing compared to PD-L1 overexpression. PD-L1-PD-1 interaction may in fact be more relevant in the context of T cell activation by APC (*Borst et al., 2021*; *Lin et al., 2018*; *Peng et al., 2020*). From the tumor side, ICAM-1 appears to be important for the physical interaction with CTL with little signaling function in this context (*Basu et al., 2016*; *Petit et al., 2016*). Indeed, expression of *ICAM1* missing the cytoplasmic domain rescued killing to the same extent as full-length ICAM-1. However, membrane self-association and possibly distribution appeared to be crucial since GPI-anchored ICAM-1, largely found as monomers in lipid rafts (*Yang et al., 2004*), did not result in productive CTL interaction. Consistent with that, dimerization and clustering of ICAM-1 is functionally important for orientation on the cell surface (*Jun et al., 2001*) and for enhancing avidity and affinity for LFA-1 binding (*Miller et al., 1995*; *Reilly et al., 1995*).

In contrast to the membrane-bound form, sICAM-1 appears to inhibit tumor cell killing (*Becker et al., 1993*). This effect was only observed in cells overexpressing sICAM-1 and not by addition of external sICAM-1 indicating a cell intrinsic mechanism leading to protection. It is possible that high local concentration at the immunological synapse is needed to prevent ICAM-1/LFA-1 interaction and induce protection from tumor killing. The pro-tumorigenic function of sICAM-1 (*Gho et al., 2001*) may explain the lack of selective pressure for ICAM-1 loss. Instead, tumor killing may be regulated by the ratio of membrane-bound vs. sICAM-1 (*Figure 8F*). The mutations Y474A, Y485A (*Tsakadze et al., 2004*), and P404E (*Fiore et al., 2002*) decreased proteolytic cleavage of ICAM-1 and subsequently shedding of ICAM-1. These results are contrary to what we found in our model indicating some cell types may employ different mechanisms to regulated ICAM-1 shedding. TCGA data analysis showed upregulation of expression of ICAM-1 cleavage related metalloproteases in different human cancers. Upregulation of *ICAM1* expression in human cancers should result in release of sICAM-1, favoring tumor growth. Furthermore, clinical data have shown that sICAM-1 is significantly upregulated in CRC patients and associated with poor prognosis (*Schellerer et al., 2019*; *de Waal et al., 2020*). A meta-analysis of 23 studies in lung cancer patients disclosed that serum sICAM-1 were significantly higher than in healthy controls and was negatively correlated with prognosis (*Wu et al., 2020*). These studies and our data strengthen the role of ICAM-1 isoforms in regulating antigen specific tumor cell killing by CTLs. Since it was recently shown that IFN-1-induced ICAM-1 expression can surmount PD-L1/PD-1 axis (*Dong et al., 2021*), increased killing could be achieved by ICB enhancing mICAM-1 expression over sICAM-1 expression. This might be achieved by using selective MMP-9, ADAM10, or ADAM17 inhibitors that prevent cleavage of ICAM-1 from the cell surface. As the expression of metalloproteases is enhanced in many cancer types and associated with tumor progression, invasion and metastases, many clinical trials of metalloprotease inhibitors have been initiated (*Das et al., 2020*; *Duffy et al., 2009*; *Gobin et al., 2019*). However, no metalloprotease inhibitor was successful so far, either due to severe side effects or lack of survival benefit (*Vandenbroucke and Libert, 2014*). Developing new, more selective, and safer metalloprotease inhibitors might circumvent these issues (*Winer et al.,*

*2018*). Alternatively, depletion of soluble ICAM-1 by pH-dependent antibodies which are selected to preferentially bind sICAM-1 proximal to the plasma membrane might be effective in removing even high levels of sICAM-1. Such 'sweeping' antibodies need not block ICAM-1 interactions as they induce lysosomal destruction of the bound sICAM-1 and have been reported for targets like C5a (100 µg/mL in plasma; *Klaus and Deshmukh, 2021*). This approach would probably be required since sICAM-1 levels are relatively high in plasma of cancer patients.

The impact of mICAM-1 and sICAM-1 on myeloid cells such as dendritic cells and macrophages and how this may affect CTL-mediated killing is not completely clear. It was previously shown that sICAM-1 stimulates the recruitment of myeloid cells (*Suarez-Carmona et al., 2015*). Membrane-associated ICAM-1 on dendritic cells facilitates T cell priming and activation leading to enhanced CTL survival and memory (*Scholer et al., 2008*), whereas the presence of a soluble form may have the opposing effect. Furthermore, ICAM-1 directly regulates macrophage polarization (*Gu et al., 2017*) affecting CTL killing of tumor cells. Further studies are necessary to investigate the role of mICAM-1 and sICAM-1 in myeloid cells and how this in turn may affect CTL-mediated killing.

# Materials and methods

## Tumor cell lines

Colon carcinoma HCT 116 (CCL-247) and pancreatic carcinoma Panc-1 (CRL-1469) cells were purchased from American Type Culture Collection (ATCC). Renal carcinoma A498 (CVCL_1056), breast carcinoma BT-549 (CVCL_1092), SF-539 (CVCL_1691), breast carcinoma MCF-7 (HTB-22), glioblastoma SNB-19 (CVCL_0535) and melanoma UACC-257 (CVCL_1779) cells were purchased from the National Cancer Institute (NCI). Breast carcinoma MDA231-TGL (CVCL_VR35) cells were purchased from European Collection of Authenticated Cell Cultures (ECACC). BT-549, MDA231-TGL, SNB-19 and SF-539 cells were cultured in Gibco RPMI-1640 Medium (Thermo Fisher Scientific, 11875093) supplemented with 10% Gibco fetal bovine serum (FBS) (Thermo Fisher Scientific, 10500064). HCT 116 cells were cultured in Gibco McCoy's 5 A Medium (Thermo Fisher Scientific, 16600082) supplemented with 10% FBS. Panc-1 cells were cultured in Gibco Dulbecco's Modified Eagle's Medium (DMEM) (Thermo Fisher Scientific, 61965026) supplemented with 10% FBS. A498 cells were cultured Eagle's Minimum Essential Medium (EMEM) (ATCC, 30–2003) supplemented with 10% FBS. MCF-7 cells were cultured in EMEM with 10% FBS and 0.01 mg/ml bovine insulin. UACC-257 cells were cultured in RPMI-1640 Medium GlutaMAX (Thermo Fisher Scientific, 61870036) supplemented with 10% FBS. A498 cells were cultured in RPMI-1640 Medium supplemented with 20% FBS. All cells were maintained at 37 °C and 5% $CO_2$ in vented flasks and splitted as recommended by the Vendor. Cell lines have been authenticated by STR profiling. Cell lines were confirmed mycoplasma negative by Mycoplasmacheck (eurofins) based on a standardized qPCR test.

## Isolation, in vitro stimulation and expansion of primary human PBMCs

Fresh blood was obtained from CMV-seropositive, HLA-A*0201[+] healthy volunteers provided by the DRK Ulm. Samples used in this study were collected from two different Donors, Donor 1 (age: 24, sex: male) and Donor 2 (age: 27, sex: male). PBMCs were isolated from heparinized fresh blood by standard density gradient centrifugation with Ficoll-Paque Plus (GE Healthcare Bio-Sciences, 17144002). PBMCs from HLA-A*0201 Donors were either stimulated with 1 µg/mL CMV pp65 antigen peptide NLVPMVATV (HLA-A*0201) (IBA Lifesciences, 6-7001-901) for 1 hr or not, washed once with medium, mixed equally and $1.5 \times 10^6$ cells/mL cultured in complete RPMI medium GlutaMAX supplemented with 10% FBS, 50 µM β-Mercaptoethanol (Thermo Fisher Scientific, 31350010) and 40 ng/mL IL-2 (BioLegend, 589102). After 4 days a half-medium change was done adding fresh complete medium and cells were further cultured for 4 days. PBMCs containing expanded antigen specific CTLs were either directly used for tumor killing assay or immediately frozen at –80 °C and thawed one day before tumor killing assay and cultured in complete medium as described above.

## CMV tetramer staining of PBMCs

For CMV-specific MHCI tetramer staining, human PBMCs ($3 \times 10^5$ cells/condition) were incubated with anti-CD8 (BD, 562428, RRID:AB_11154035), anti-CD25 (BD, 564467), anti-PD-1 (BD, 561272, RRID:AB_2744340), anti-LAG-3 (BioLegend, 369212, RRID:AB_2728373) or anti-LFA-1 (BD, 559875,

RRID:AB_2129113) antibodies or respective isotype control antibodies where indicated and PE-CMV tetramer (MBL International, TB-0010–1) or PE-control tetramer (MBL International, TB-0029–1) in FACS buffer containing 1% human Fc Block (Miltenyi Biotec, 130-059-901, RRID:AB_2892112) for 30 min at 4 °C and were then washed three times. Flow cytometry analyses were performed using LSRFortessa (BD Biosciences) and data were analyzed using FlowJo version 10.8 (FlowJo LLC).

## Tumor killing assay

For the tumor killing assay, HLA-A*0201 positive tumor cells were used as target cells. Tumor cells were either kept untreated or were incubated with CMV pp65 antigen peptide (IBA Lifesciences, 6-7001-901) for 1 hr at 37 °C and washed once with medium. Untreated or antigen loaded tumor cells were seeded in 96-well plates and allowed to attach for 1–2 hr before PBMCs containing antigen specific expanded CTLs were added in different target to effector (T:E) ratios in triplicate. In experiments where recombinant sICAM1 function was assessed, human sICAM-1 (Thermo Fisher Scientific, BMS313) or human ICAM-1 comprising the extracellular domain (Preprotech, 150–05) was added at the beginning of co-culture or in regularly-spaced intervals. After 3 days of co-culture, the viability of cells was assessed using CellTiter-Glo reagents (Promega, G7571) according to the manufacturer's protocol. The survival of target cells for each T:E was calculated using GraphPad Prism as percentage of target cell survival normalized to values obtained from untreated tumor cells not incubated with PBMCs. Respective values of PBMCs only or medium (blank) were subtracted from obtained raw values. To measure real-time kinetic of tumor cell killing, tumor cells transfected with plasmids containing eGFP were treated and co-cultured as described above in the IncuCyte SC5 Live-Cell Analysis system (Sartorius). Plates were scanned with a 10 x objective using phase contrast channel as well as the green fluorescence channel for 42 hr every 6 hr. Data were analyzed by counting green objects over time and normalized to t=0 hr to determine survival of transfected tumor cells. Conditions were performed in triplicate and 4 pictures of each triplicate were used for analysis (in total 12).

## Generation of Cas9 and dCas9 stable tumor cell lines

Lentiviral hEF1α-Blast-Cas9 Nuclease (Dharmacon, VCAS10126) and hEF1a-Blast-dCas9-VPR Nuclease (Dharmacon, VCAS11922) was used to transduce HCT 116 and Panc-1 cells with a MOI of 0.3. Single cell clones of transduced cell lines were obtained by limiting dilution and clonal expansion. Transduced cells were selected with 10 µg/mL Blastidicin S HCl (Thermo Fisher Scientific, A1113903). Best single cell clones for each cell line were chosen based on expressed amount of Cas9/dCas9 protein and editing efficiency.

## Activation of gene expression using CRISPRa

A total of 2x10^5 HCT 116 cells stably expressing dCas9 per well were seeded in 6-well plates one day before transfection. Cells were transfected using DharmaFECT 4 reagent (Horizon Discovery, T-2004–02) according to manufacturer's instructions. Briefly, 25 nM crRNA pool targeting *CD274*, *CD80*, *NT5E* (see *Table 1*) or a non-targeting control (NTC) (Horizon Discovery, U-009500-10-05) were mixed equal with 25 nM tracrRNA (Horizon Discovery, U-002005–50) in serum-free medium (Thermo Fisher Scientific, 31985062). DharmaFECT transfection reagent was diluted 1:50 in serum free medium und mixed

**Table 1.** crRNA sequences used for inducing gene expression through CRISPRa.

| Target gene | Pool crRNA sequences | Company | Catalogue nr. |
|---|---|---|---|
| *CD274* | TCGGCGGAAGCTTTCAGTTT, GCTTCCGCCGATTTCACCGA, CGTTGCGCCAGGCCCGGAGG, CAGCGTTGCGCCAGGCCCGG | Horizon Discovery | P-015836-01-0005 |
| *CD80* | CCACGAGCACCAGGCGGCCT, TAGTCCATGCACGGTGGTGA, GTCAGTGCCAGGAGTTGGAC, AATGGTGCCCGAGAAGAGTG | Horizon Discovery | P-007851-01-0005 |
| *NT5E* | TCCGGGTACCAGGTCGGAT, TCCGACCCTGGTACCCGGAG, CAGGGCCGCTCCGGGTACCA, GACGTCACCCGATCCGACCC | Horizon Discovery | P-008217-01-0005 |

1:1 with crRNA:tacrRNA working solution and incubated for 20 min. After adding 1600 µL growth medium to the transfection crRNA:tacrRNA mix, growth medium in six-well plate was removed and transfection mix was added to each well. Gene expression for each gene was measured over time using flow cytometry. Shortly, cells were harvested at different time points and $1 \times 10^5$ cells were stained with anti-CD274 (BioLegend, 329705, RRID:AB_940366), anti-CD80 (BD, 564159, RRID:AB_2738631), anti-NT5E (BioLegend, 344003, RRID:AB_1877224), FVS520 (BD, 564407, RRID:AB_2869573) or FVS660 (BD, 564405, RRID:AB_2869571) and respective isotype controls in FACS buffer containing 1% Fc-Block for 30 min at 4 °C. Cells were washed three times and analyzed by flow cytometry.

## Construction of sgRNA libraries
The CRISPR KO library consisting of 64,556 human sgRNA sequences (6 sgRNAs/gene) was designed according to the Vienna Bioactivity CRISPR score (VBC score) (*Michlits et al., 2020*). The CRISPRa library consisting of 67,832 sgRNA (6 sgRNAs/gene) sequences was designed based on the Weissmann CRISPRa library V2 (*Horlbeck et al., 2016*). The sgRNA sequences were synthesized by Twist Biosciences and cloned into a lentiviral sgRNA expression vector pLenti-sgETN as described in *Lindner et al., 2021* (pLenti-U6-sgRNA-EF1as-Thy1.1_P2A_NeoR) (*Lindner et al., 2021*).

## Lentivirus production and purification
For lentivirus production, the Lenti-X 293T cell line (Takara, 632180) was used. Cells were seeded on Collagen I coated culture dishes (Biocoat, 356450) in DMEM supplemented with 10% FBS to be 70–80% confluent. After 6 hr, cells were transfected with a mixture of PEI, KO/activation sgRNA library pools and MISSION lentiviral packaging mix (Sigma, SHP001) in serum free Opti-MEM media (Thermo Fisher Scientific, 31985062). Before transfection, the mix was incubated for 20 min at RT followed by dropwise addition to the cells. On the next day, transfection media was replaced by new DMEM supplemented with 10% FBS. Virus containing media was harvested 48 hr and 72 hr post transfection and pooled. Cell debris was removed by centrifugation at 3000 *g* for 15 min. Media containing virus particles was mixed with PEG-it virus precipitation solution (System Biosciences, LV810A-1) and incubated at 4 °C overnight. Viral supernatants were centrifugated at 1500 *g* for 30 min at 4 °C and obtained virus pellets were resuspended in resuspension buffer and subsequently frozen in aliquots at − 80 °C. Virus quantification of KO/activation pool was done by droplet digital PCR (ddPCR) using QX200 Droplet Digital PCR System (Bio-RAD, 1864001).

## CRISPR screens and genomic DNA extraction
CRISPRa and CRISPR KO screen were performed using HCT 116 dCas9 and HCT 116 Cas9 cells. Cells were transduced with sgRNA KO library or sgRNA activation library, respectively, and selected with 800 µg/mL G418 (Invitrogen, 10131035) for 8 days. The transduced cells were cultured with three different conditions: (1) tumor cells loaded with antigen or (2) not and co-cultured with PBMCs containing expanded antigen specific CTLs, and (3) untreated tumor cells alone as control group. For the CRISPR KO screen, a tumor cell:PBMC ratio of 2:1 was used whereas for the CRISPRa screen a ratio of 1:2 was selected. After a co-culture phase of 3 days, dead tumor cells and PBMCs were washed away with PBS (Thermo Fisher Scientific, 0010056) and remaining living tumor cells were harvested using TrypLE Select Enzyme (1 X; Thermo Fisher Scientific, 10010023) and counted to determine amount of killed tumor cells. To access sgRNA library representation genomic DNA was isolated from remaining tumor cells. First, cells were digested with Proteinase K solution (Thermo Fisher Scientific, 25530049) for 24 hr and subsequently heat-inactivated at 95 °C for 10 min. Followed by RNase A (Qiagen, 19101) digestion for 30 min and homogenization using QIAshredder (Qiagen, 79654). DNA was extracted by using ROTIPhenol/Chloroform-Isoamylalkohol (Roth, A156.3), precipitated and washed with Ethanol (Honeywell Research Chemicals, 32205) and finally centrifuged. Each DNA pellet was resuspended in 150 µL elution buffer (Qiagen, 1014819).

## CRISPR screens readout
To determine sgRNA abundance as screen readout, initial PCR amplification of sgRNA cassettes adding overhang adapter sequence was performed using Q5 Hot Start High-Fidelity 2 X Master Mix (New England Biolabs, M0494S). For each sample, 1 µg extracted genomic DNA was used in a 100 µL reaction run with the following cycling conditions: 98 °C for 1 min, 25 cycles of (98 °C for 15 s, 55 °C

**Table 2.** sgRNA sequences used to knockout *ILKAP* and *ICAM1* for validation experiments.

| Target gene | sgRNA Name | sgRNA sequence | Thermo Fisher Scientific Identifier | Catalogue nr. |
|---|---|---|---|---|
| *ILKAP* | ILKAP sgRNA1 | TTCGGTGATCTTTGGTCTGA | CRISPR617045_SGM | A35533 |
| *ILKAP* | ILKAP sgRNA2 | GATGTCGTTCAGGATGACGT | CRISPR617051_SGM | A35533 |
| *ILKAP* | ILKAP sgRNA3 | GCCATTCTTCTCTTCCTCGG | CRISPR617058_SGM | A35533 |
| *ICAM1* | ICAM1 sgRNA1 | GGTCTCTATGCCCAACAACT | CRISPR845341_SGM | A35533 |
| *ICAM1* | ICAM1 sgRNA2 | GCTATTCAAACTGCCCTGAT | CRISPR845351_SGM | A35533 |
| - | Non-targeting control (NTC) | - | | A35526 |

for 30 s, 72 °C for 30 s), and 72 °C for 2 min. Pooled PCR products from each sample were purified using Agencourt AMPure XP (Beckman Coulter, A63880) with a PCR-product/bead ratio of 1:0.8. In a second PCR, purified PCR products were amplified using indexed adapter primers from Illumina to generate barcoded amplicons and NEBNext Ultra II Q5 Master Mix (New England Biolabs, M0544S). For each index PCR, 20 ng template was used in a 50 µL reaction with following cycling conditions: 98 °C for 30 min, 7 cycles of (98 °C for 10 s, 65 °C for 75 s), and 65 °C for 5 min. Index-PCR products were purified twice as described before and eluted in 30 µL. For Next-Generation Sequencing, all library samples were pooled, diluted, 10% PhiX was added and then sequenced with NextSeq 500/550 High Output Kit v2.5 (Illumina, 20024907).

## Generation of *ILKAP* KO and *ICAM1* KO cells

For gene hit validation experiments, KO cell lines were generated using the CRISPR-Cas9 system. To generate bulk cell pools, HCT 116 Cas9 and Panc-1 Cas9 cells were transfected with two to three independent sgRNAs targeting *ILKAP* (see *Table 2*) using DharmaFECT 4 Transfection reagent (Horizon Discovery, T-2004–03). according to manufacturer's instructions. After 2 days, cells were used for tumor killing assay and Simple Western analysis. Limiting dilution and clonal expansion was used to generate HCT 116 *ILKAP* KO monoclonal cell pools for further analysis. Gene disruptions were confirmed by sequence analysis and Simple Western analysis. To generate *ICAM1* KO polyclonal cell pools, HCT 116 Cas9 and Panc-1 Cas9 were transfected with two independent sgRNAs targeting *ICAM1* (see *Table 3*) using DharmaFECT 4 Transfection reagent (Horizon Discovery, T-2004–03) according to manufacturer's instructions. UACC-257 cells were co-transfected with Cas9 protein and two independent sgRNAs targeting *ICAM1* using Lipofectamine CRISPRMAX Cas9 Transfection Reagent (Thermo Fisher Scientific, CMAX00008) according to manufacturer's instructions. ICAM-1 negative cells were sorted using fluorescence-activated cell sorting (FACS) and further expanded, then used for tumor killing assay and validation experiments. Depletion of *ICAM1* was periodically checked by cell surface staining.

## Protein analysis using western blot/simple western system

Cells were collected for immunoblotting analysis, washed with 1 x PBS and lysed with RIPA buffer (Thermo Fisher Scientific, 89901) supplemented with protease inhibitors (Thermo Fisher Scientific,

**Table 3.** Overview of libraries used for comparisons in each biological contrast.
PBMC - Peripheral Blood Monocyte Cells, RRA - Robust Rank Aggregation, MLE - Maximum Likelihood Estimation.

| Biological contrast | Control | Treatment | CRISPR-screen hits identification method | FDR cutoff [%] |
|---|---|---|---|---|
| Tumor screen | Plasmid gRNAs libraries | Only tumor cells without PBMC | RRA | 2 |
| Antigen-independent tumor killing | Only tumor cells without PBMC | Tumor cells not loaded with antigen with PBMC | MLE | 5 |
| Antigen-dependent tumor killing | Tumor cells not loaded with antigen with PBMC | Tumor cells loaded with antigen with PBMC | MLE | 5 |

78329) for 30 min at 4 °C. After incubation, lysates were centrifuged at 16,000 $g$ for 10 min at 4 °C and supernatants were collected in new tubes. Protein quantification was done by using Pierce BCA Protein Assay Kit (Thermo Fisher Scientific, 23225). For traditional immunoblotting, samples were diluted to a 1 µg/µL protein concentration with RIPA buffer, mixed with LDS sample buffer (Thermo Fisher Scientific, NP0007) supplemented with Sample Reducing Agent (Thermo Fisher Scientific, B0009) and boiled at 95 °C for 5 min. Twenty µL of each sample were loaded in 4 bis 12%, NuPAGE Bis-Tris gels (Thermo Fisher Scientific, NP0335BOX) and ran for 40 min at 200 V. After electrophoresis, protein was transferred to a nitrocellulose membrane for 7 min at 25 V using the iBlot 2 Dry System (Thermo Fisher Scientific, IB21001). Membranes were incubated for 1 hr in RotiBlock solution (Roth, Cat#A151.2), and then overnight at 4 °C with Anti-ILKAP (Thermo Fisher Scientific, PA5-52100, RRID:AB_2642706) and Anti-αTubulin (Cell Signaling, Cat#2144 S) antibodies diluted 1:500 in blocking solution. The next day, membranes were washed three times with TBST (Roth, Cat#1061.1) washing solution. Then, HRP-tagged Anti-mouse (Thermo Fisher Scientific, Cat#31430) and rabbit (Thermo Fisher Scientific, Cat#31460) 2[ary] antibody antibodies were added at 1:10.000 dilution, followed by three more steps of washing. Membranes were incubated in SuperSignalTM West Pico PLUS Chemiluminescent Substrate (Thermo Scientific, Cat#34580) and signal was detected for 10 s in ImageQuant LAS 4000 (Cytiva). Protein analysis was also performed using the Protein Simple Western/Peggy Sue platform (Bio-Techne), a capillary electrophoresis immunoassay. Protein samples were in this case diluted in 0.1 X sample buffer 2 (Protein Simple) to a concentration of 0.05 mg/ml. Anti-ILKAP and anti-β-actin (Sigma Aldrich, A5441, RRID:AB_476744) antibodies were used at a 1:10,000 and 1:25 dilution, respectively, and ran according to the manufacturer's instructions. Data were analyzed with Compass software (Compass for SW Version 5.0.0). The peak area values of each sample were normalized to β-actin. Data from three independent runs were pooled and analyzed in GraphPad Prism.

## Production of sICAM-1 conditioned medium

A total of $3.5 \times 10^6$ WT or ICAM1 KO Panc-1 cells were respectively seeded in 10 mL of Panc-1 medium enriched with 20 ng/mL recombinant human TNFα (Biolegend, Cat#570106) in a T75 flask. Twenty-four hr after, the full volume was exchanged with 6 mL of fresh assay medium to remove excess TNFα. Cells were then further incubated for 48 hr, after which supernatant was collected, centrifuged at 1000 $g$ for removal of cellular debris and stored for later usage.

## Treatments of tumor cells

A total of $1.5 \times 10^4$ cells per well were seeded in 96-well plates with medium containing either 100 ng/mL IFN-γ (Bio Legend, 570202) or 40 ng/mL TNF-α (Bio Legend, 570104) for two days. Cells were harvested and incubated with conjugated monoclonal anti-HLA-A2 (BioLegend, 343306, RRID:AB_1877227), anti-PD-L1 (Bio Legend, 329713, RRID:AB_10901164) and anti-ICAM-1 (BD, 559771, RRID:AB_398667) antibodies or respective isotype for 30 min at 4 °C. Nonspecific binding was blocked by using 1% Fc block. Cells were washed three times and analyzed by flow cytometry. Cell viability was determined using fixable viability stain FVS780 (BD, 565388, RRID:AB_2869673).

## Design, transfection and detection of ICAM-1 variants containing eGFP-plasmids

Sequences for full-length ICAM-1 and isoforms were obtained from Uniprot, n-terminal Flag-tag was added and optimized for expression in humans by GeneArt Optimization. Then it was cloned by GeneArt into an Boehringer Ingelheim inhouse vector (pOptiVec-Blast-eGFP). For validation experiments, $2 \times 10^5$ cells per well were seeded in six-well plates 1 day before transfection of constructed plasmids. Cells were transfected using Invitrogen Lipofectamine 3000 (Thermo Fisher Scientific, L3000015) according to manufacturer's instructions. After 1 day, transfected cells were harvest for real-time tumor killing assay and flow cytometry. Additionally, supernatants were collected for IQELISA analysis. For flow cytometry, $1.5 \times 10^5$ cells were stained with anti-DYKDDDK(Flag)-tag antibody (BioLegend, 637315, RRID:AB_2716154) or isotype control for 30 min at 4 °C, washed three times and analyzed by flow cytometry.

## Detection of sICAM-1

The amount of sICAM-1 in harvested cell culture supernatants was measured by using either RayBio human sICAM-1 IQELISA kit (RayBiotech, IQH-ICAM1) or human sICAM-1 ELISA kit (RayBiotech, ELH-ICAM-1) according to manufacturer's instructions in duplicates for each sample. IQELISA readout was done with a Quantstudio 6 Flex system (Life Technologies Corporation) and raw data were analyzed by Quantstudio Real-Time PCR System v.1.7.1 (Life Technologies Corporation). Concentrations of sICAM-1 were quantified by interpolation from the standard curve using GraphPad prism software and fold change was calculated.

## Statistical analysis

Graphs and statistical analysis was done using Prism version 9 (GraphPad) as indicated in the figure legends. In general, data between two groups were compared using a two-tailed unpaired Student's *t* test. To compare multiple groups multiple unpaired *t* tests with adjustments for multiple comparisons was performed. To compare multiple groups to a control group an analysis of variance (ANOVA) for multiple comparison according to Dunnett was used. Statistical significance is displayed on the figures with asterisks as follows: *, $p<0.05$; **, $p<0.01$; ***, $p<0.001$; ****, $p<0.0001$; $p>0.05$ was considered not significant. The number of technical or biological replicates (n value; independent experiments) is indicated for each figure. Throughout the manuscript, no power analysis was used, but group size was based on previous studies using comparable approaches.

## Computational methods

### Reads processing

CRISPR-Cas9 libraries were single read sequenced in two separate batches:(1) plasmid libraries and (2) tumor killing screens. Acquired reads were trimmed using cutadapt (*Martin, 2011*) v1.8.1 with the following options: *-n 1 --match-read-wildcards --trimmed-only --minimum-length 17* using the following adapter sequences: 3': CTTGTGGAAAGGACGAAACACC and 5': GTTTAAGAGCTATGCT GGAAACAGCATAG. Trimmed reads were aligned to the gRNA and respective target genes, counted and scored using MAGeCK-VISPR v.0.5.3 (*Li et al., 2015*) using the human genome version hg38 and other default options.

### Identifying CRISPR screen hits

The significant screen hits in respective biological contrasts were determined by comparing control against treatment libraries using methods and conditions described in *Table 3*.

---

**Table 4.** General design matrix for MLE comparison for specificity of antigen in- and dependent CRISPR-Cas9 screens.

TC - tumor cells; PBMC - co-culture with PBMC or lack of it (noPBMC), AG - PBMC antigen stimulation or lack of it (noAG); rep1,2,3 – technical replicates.

| Samples | Baseline | Antigen independent | Antigen dependent |
|---|---|---|---|
| TC_noPBMC_noAG_rep1 | 1 | 0 | 0 |
| TC_noPBMC_noAG_rep2 | 1 | 0 | 0 |
| TC_noPBMC_noAG_rep3 | 1 | 0 | 0 |
| TC_PBMC_noAG_rep1 | 1 | 1 | 0 |
| TC_PBMC_noAG_rep2 | 1 | 1 | 0 |
| TC_PBMC_noAG_rep3 | 1 | 1 | 0 |
| TC_PBMC_AG_rep1 | 1 | 0 | 1 |
| TC_PBMC_AG_rep2 | 1 | 0 | 1 |
| TC_PBMC_AG_rep3 | 1 | 0 | 1 |

---

## CRISPR screen hits evaluation

The screen hits were intersected with the common essential genes (*Tsherniak et al., 2017*) provided by *DepMap, 2020* Q4 version (*DepMap, 2020*). Additionally, CRISPR screen hits were intersected with the consensus core set of 182 genes from CRISPR-Cas9 screened mouse models published (*Lawson et al., 2020*). Mouse gene symbols were translated into human orthologs (one-to-one) using biomart, highly confident annotation (*Kinsella et al., 2011*) which resulted in 162 orthologs.

## Specificity of biological contrast hits

Specificity of antigen-dependent and independent hits in each of the screen types (KO or activation) was determined using the double contrast MLE approach implemented in MAGeCK-VISPR (*Li et al., 2015*) and the design matrix in *Table 4* was used for the comparison. All resulting β-scores were normalized for cell-cycle differences between the cell cultures using the normalization feature implemented in MAGeCK-FLUTE (*Wang et al., 2019*). The target gene was considered as hit either in activation or in KO, or common if it was a hit in both screens, in which β-score absolute value was higher than 1 and FDR-corrected Wald's test p-value was less than 0.05. Similarly, the gene was contrast-specific if it was a hit in any of the considered screens. All the genes that did not pass any of the described criteria were considered not significant.

## Screen hits correlation

The correlation coefficient (Pearson's or Spearman's) between the CRISPRa and CRISPR KO screen hits was performed in the signaling pathway-specific manner using the base R cor function (*R Development Core Team, 2022*). Firstly, in the CRISPRa and CRISPR KO screen, MAGeCK calculated scores were quantile normalized with the limma R package (*Ritchie et al., 2015*). All genes were assigned to KEGG pathways using KEGG REST (*Tenenbaum, 2020*) and MetaCore annotations (*Analytics, 2021*). Finally, the correlation coefficient between quantile-normalized scores was calculated for the genes that were considered a hit in either CRISPRa or CRISPR KO screen within each signaling pathway. Fisher's exact test was calculated in a signaling pathway-specific manner using the stats R package (*Vahedi et al., 2012*) and the following contingency table: CRISPRa and CRISPR KO against screen hit or not a hit.

## Functional analysis

Gene ontology (GO) and signaling pathway enrichment analysis was performed using g:Profiler *Raudvere et al., 2019* for human annotation and a union of all CRISPR-Cas9 targeted genes was used as the gene universe. All results were multiple test corrected (FDR - correction) and only the terms or pathways with adjusted p-value of less than 0.05 were considered. GO terms were clustered according to their semantic similarity using Wang's distance (*Wang et al., 2007*) and implemented in the rrvgo R package (*Sayols, 2020*). Briefly, all enriched GO terms were pooled and each of them was assigned a score equal to its -log10 adjusted p-value. The terms were hierarchically clustered (complete linkage method) with a threshold of 0.9 and a single representative of each of the top 40 scoring, non-redundant clusters was used for results visualization.

## Visualization and plotting of CRISPR screen data

All graphs were plotted using ggplot2 (*Bowes et al., 2016*) and combined with patchwork (*Pedersen, 2020*). The upset plots were generated using the UpSetR R (*Conway et al., 2017*). Circular chromosome plot was generated using RCircos (Version 1.2.1) R package (*Zhang et al., 2013*).

## Survival analysis

The patients' clinical data from TCGA and GTEx for the following cancer types: colorectal adenocarcinoma, breast carcinoma, breast invasive carcinoma, head and neck squamous cell carcinoma, hepatocellular carcinoma, glioblastoma multiforme, lung adenocarcinoma, pancreatic adenocarcinoma, skin cutaneous melanoma, and gastrointestinal tumor, were split into three groups for each enquired

gene. Each data point was classified as: low, medium, and high if the selected gene's expression was respectively below 25th, between 25th and 75th, and above 75th percentile in a given patient sample. The reference group for the two genes survival analysis was set to high-high. The differences between the groups were tested using Cox proportional hazard model (*Therneau and Grambsch, 2000*) implemented in the survival R package (*Therneau, 2022*). The Kaplan-Meier plots were generated using survminer R package (*Alaterre et al., 2021*). The expression analysis and respective plots were obtained using GEPIA (*Tang et al., 2017*).

## Acknowledgements

The authors thank Verena Mücke for technical support and generating and providing B2M KO tumor cells. Additionally, we want to thank the FACS Unit especially Daniela Reiss for the FACS sorting support. The data used for the analyses described in this manuscript were obtained from the GTEx Portal. The Genotype-Tissue Expression (GTEx) Project was supported by the Common Fund of the Office of the Director of the National Institutes of Health, and by NCI, NHGRI, NHLBI, NIDA, NIMH, and NINDS. The results survival analysis are in whole or part based upon data generated by the TCGA Research Network: https://www.cancer.gov/tcga. We would like to acknowledge Venu Thatikonda for the support on multigene survival analysis. Figures were created with BioRender.com. Funding. This study was funded by Boehringer Ingelheim.

## Additional information

### Competing interests

Ann-Kathrin Herzfeldt, Marta Puig Gamez, Eva Martin, Praveen Baskaran, Heinrich J Huber, Michael Schuler, John E Park, Lee Kim Swee: was an employee at this time of Boehringer Ingelheim Pharma GmbH Co. KG. The author has no other relevant affiliations or financial involvement with any organization or entity with a financial interest in or financial conflict with the subject matter or materials discussed in the manuscript apart from those disclosed. Lukasz Miloslaw Boryn: was an Ardigen S.A. employee. The funder provided support in the form of salaries for the authors.

### Funding

| Funder | Grant reference number | Author |
|---|---|---|
| Boehringer Ingelheim | | Ann-Kathrin Herzfeldt<br>Marta Puig Gamez<br>Eva Martin<br>Lukasz Miloslaw Boryn<br>Praveen Baskaran<br>Heinrich J Huber<br>Michael Schuler<br>John E Park<br>Lee Kim Swee |

The funders had no role in study design, data collection and interpretation, or the decision to submit the work for publication.

### Author contributions

Ann-Kathrin Herzfeldt, Conceptualization, Formal analysis, Investigation, Methodology, Validation, Visualization, Writing – original draft, Writing – review and editing; Marta Puig Gamez, Eva Martin, Formal analysis, Investigation; Lukasz Miloslaw Boryn, Praveen Baskaran, Heinrich J Huber, Formal analysis; Michael Schuler, Conceptualization; John E Park, Conceptualization, Resources, Writing – original draft, Writing – review and editing; Lee Kim Swee, Conceptualization, Supervision, Writing – review and editing

### Author ORCIDs

Ann-Kathrin Herzfeldt https://orcid.org/0000-0002-7882-8284
Marta Puig Gamez https://orcid.org/0009-0006-6047-8200
Lukasz Miloslaw Boryn https://orcid.org/0000-0001-8091-1071

Praveen Baskaran (iD) https://orcid.org/0000-0003-2275-3516
Heinrich J Huber (iD) https://orcid.org/0000-0003-4454-2971
John E Park (iD) https://orcid.org/0000-0002-5674-6026

### Decision letter and Author response

Decision letter https://doi.org/10.7554/eLife.84314.sa1
Author response https://doi.org/10.7554/eLife.84314.sa2

## Additional files

### Supplementary files

- MDAR checklist

### Data availability

All data generated or analyzed are included in the manuscript. Source data files are provided for figure 2 and figure 3.

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
