## [Editor Report]

This important study uses complementary cutting-edge CRISPR approaches (CRISPR and CRISPRa) to identify novel determinants of cytotoxic CD8 T cell (CTL)-mediated tumor cell killing in vitro. The Authors use these screens to identify that the integrin-linked kinase ILKAP and the integrin protein ICAM1 both mediate resistance to CTL-mediated killing, leading to a new understanding of how some tumours may evade killing by T cells. The strength of the evidence for these findings is exceptional and backed up by the study of several cancer cell lines as well as human data. This work will be of great interest to tumor immunologists as well as those studying evasion of checkpoint therapy in cancer treatment.

---

## [Decision Letter]

**Decision letter after peer review:**

Thank you for submitting your article "Complementary CRISPR screen highlights the contrasting role of membrane-bound and soluble ICAM-1 in regulating antigen specific tumor cell killing by cytotoxic T cells" for consideration by *eLife*. We apologize for the delay in getting you these comments and list of revisions. Your article has been reviewed by 2 peer reviewers, including Brian Altman as Reviewing Editor and Reviewer #1, and the evaluation has been overseen by Kimryn Rathmell as the Senior Editor.

The reviewers have discussed their reviews with one another, and the Reviewing Editor has drafted this Consensus Essential Revisions list to help you prepare a revised submission. Revisions that require additional experimentation are listed as Major Points.

Essential revisions:

1. The Authors propose that sICAM1 is important as a secreted factor to antagonize LFA1 and decrease CTL killing, but do not present data to disentangle sICAM1 as a secreted factor and sICAM1's role in the tumor cells when it is overexpressed. To resolve this, the Authors could use conditioned media with sICAM1 to show that it decreases CTL killing of untreated cells. They could also use microscopy and measure T cell and tumor cell interaction duration and frequency under conditions with and without soluble ICAM1 present.

2. The connection between ILKAP and ICAM1 is somewhat poorly defined and needs to be better described. It is not clear from the data how ILKAP regulates ICAM1 expression, nor do the Authors show if ILKAP regulates the ratio of mICAM1 / sICAM1. Furthermore, the Authors suggest in Figure 4 that the ability of ILKAP to regulate CTL killing is not dependent on TNF or IFN, and yet in Figure 8, the Authors show that both forms of ICAM1 are induced by these same cytokines. This calls into question whether ILKAP is truly upstream of ICAM1, and what the functional significance of this connection is.

3. Regarding the analysis of clinical relevance, the authors show that patients with high levels of ICAM1 expression in combination with high levels of protease expression have poor survival. The rationale behind this is that the proteases cleave ICAM1 off the membrane leading to high levels of soluble ICAM1 that then negatively affects T cell-mediate tumor cell lysis. To demonstrate that indeed the combination of both factors, ICAM1 expression, and protease expression (MMP9/ADAM10/ADAM17), is responsible for poor survival, the authors should also have analyzed the impact of each of these factors alone on patient survival. If their hypothesis is true, the combination of high ICAM1 and protease expression should have a worse impact on survival than each factor alone. The Authors should show Kaplan Meier plots of ICAM1 and MMP9/ADAM10/ADAM17 separately along with being grouped together, to see if the expression of these genes independently correlates with survival outside of their association with each other.

---

## [Author Response]

Essential revisions:1. The Authors propose that sICAM1 is important as a secreted factor to antagonize LFA1 and decrease CTL killing, but do not present data to disentangle sICAM1 as a secreted factor and sICAM1's role in the tumor cells when it is overexpressed. To resolve this, the Authors could use conditioned media with sICAM1 to show that it decreases CTL killing of untreated cells. They could also use microscopy and measure T cell and tumor cell interaction duration and frequency under conditions with and without soluble ICAM1 present.

Indeed, additional experiments were necessary to understand the cell intrinsic vs extrinsic role of sICAM-1 in the interaction of tumor and CTL i.e. is the presence or addition of sICAM-1 sufficient to protect surrounding cells?

We already clearly demonstrated that overexpression of sICAM-1 in tumor cells leads to diminished antigen specific CTL mediated killing (Figure 6B). As suggested, to examine the effect of cell-extrinsic, supplemented sICAM-1 on tumor cell killing by CTLs, we investigated whether the addition of either conditioned media from cells secreting sICAM-1 or recombinant sICAM-1 could decrease antigen-dependent killing of tumor cells by CTL. Neither the addition of conditioned medium (Figure 6 —figure supplement Figure 2A and B) nor recombinant sICAM-1 decreased CTL killing (Figure 6 —figure supplement Figure 3A and B).

To exclude that sICAM-1 overexpression leads to reduced membrane ICAM-1, we monitored level of mICAM-1 upon sICAM-1 expression and demonstrate that mICAM-1 level do not change upon sICAM-1 overexpression and cannot explain differences in CTL killing (Figure 6 —figure supplement Figure 1).

Altogether, these results suggest that cells must produce sICAM-1 to be protected. We hypothesize that increased local concentration e.g. at the immunological synapse may be important to induce protection from CTL killing. It is possible that extrinsically added sICAM-1 does not achieve such local concentration (Grima, 2009). Based on the new findings, we adjusted our interpretation in the revised results (lines 468-489) and discussion part (lines 684-688).

2. The connection between ILKAP and ICAM1 is somewhat poorly defined and needs to be better described. It is not clear from the data how ILKAP regulates ICAM1 expression, nor do the Authors show if ILKAP regulates the ratio of mICAM1 / sICAM1. Furthermore, the Authors suggest in Figure 4 that the ability of ILKAP to regulate CTL killing is not dependent on TNF or IFN, and yet in Figure 8, the Authors show that both forms of ICAM1 are induced by these same cytokines. This calls into question whether ILKAP is truly upstream of ICAM1, and what the functional significance of this connection is.

To clarify the connection between ILKAP and ICAM1, we conducted further experiments. We overexpressed ILKAP and a catalytic inactive mutant of ILKAP (H154D) (Leung‐Hagesteijn et al., 2001) in tumor cells and assessed levels of mICAM-1 on the cell surface and sICAM-1 in the supernatant. In Panc-1 cells, we observed a consistent trend toward lower mICAM-1 and sICAM-1 levels following transient ILKAP but not catalytic dead ILKAP (H154D) transfection, however, the results were not statistically significant (Figure 8, H and I) (lines 588-595). The minor impact of overexpression compared to deletion could be explained by already high level of ILKAP phosphatase activity in untransfected cells that may not be dramatically increased upon overexpression or regulation of ILKAP levels by cells. It was previously shown that Integrin linked kinase (ILK) regulates ICAM-1 expression among others via NF-κB signalling (Lee et al., 2006). The catalytic inactive mutant of ILKAP (H154D) do not inhibit ILK kinase activity (Leung‐Hagesteijn et al., 2001) and no changes in ICAM-1 levels in cells overexpressing this mutant could be detected in our experiment. Thus, we propose that ILKAP may controls ICAM-1 levels through modulating ILK signalling.

With respect to IFN-γ, TNFα, ICAM-1 and ILKAP, our data show that:

1) ILKAP deficiency does not increase sensitivity to IFN-γ as it is the case for other phosphatases such as PTPN2

2) ILKAP regulate ICAM-1 expression, most likely independently of IFN-γ/TNFα stimulation since KO or overexpression influenced mICAM-1 level

3) IFN-γ and TNFα up-regulate ICAM-1, apparently independently of ILKAP

Altogether, ILKAP and ICAM1 signals appear to be connected through another, yet to be identified, component, which, however, is not IFN-γ or TNFα.

3. Regarding the analysis of clinical relevance, the authors show that patients with high levels of ICAM1 expression in combination with high levels of protease expression have poor survival. The rationale behind this is that the proteases cleave ICAM1 off the membrane leading to high levels of soluble ICAM1 that then negatively affects T cell-mediate tumor cell lysis. To demonstrate that indeed the combination of both factors, ICAM1 expression, and protease expression (MMP9/ADAM10/ADAM17), is responsible for poor survival, the authors should also have analyzed the impact of each of these factors alone on patient survival. If their hypothesis is true, the combination of high ICAM1 and protease expression should have a worse impact on survival than each factor alone. The Authors should show Kaplan Meier plots of ICAM1 and MMP9/ADAM10/ADAM17 separately along with being grouped together, to see if the expression of these genes independently correlates with survival outside of their association with each other.

We thank the reviewer for bringing up this interesting point. We examined the impact of *ICAM1* expression and protease expression on patients’ survival separately. The Kaplan Meier plots were included in Figure 7 —figure supplement Figure 1. The impact of the expression of the combination of high ICAM1 and protease expression have a worse impact on survival than each gene alone (lines 538-539).